# Quantitative live imaging reveals PRICKLE1 controls junctional neural tube morphogenesis independent of Planar Cell Polarity

Jian Xiong Wang ©[1], Yanina D. Alvarez[1], Siew Zhuan Tan ©[1], Samara N. Ranie[1], Samantha J. Stehbens ©[1,2,3] & Melanie D. White ©[1,4] ✉

Neurulation, the process that forms the neural tube - the precursor to the brain and spinal cord - is frequently disrupted in congenital malformations. Primary and secondary neurulation are integrated at a junctional zone, yet the cellular dynamics linking these programs remain unknown. Using high-resolution quantitative live imaging in transgenic quail embryos, we show that the junctional neural tube forms through two coordinated processes: mediolateral convergence and EMT-driven ingression of medial neuroepithelial cells. We demonstrate that PRICKLE1, a core PCP protein, orchestrates these behaviors independently of planar polarity cues. PK1 is enriched at the apical cortex of medial cells, where it drives actomyosin accumulation and apical constriction. This function is essential for both convergence and cell ingression but is uncoupled from classical PCP axis establishment. Our findings redefine the molecular basis of junctional neurulation and implicate impaired EMT as a central cause of localized neural tube defects.

Neurulation is a critical early embryonic process in vertebrates which forms the neural tube, the precursor to the brain and spinal cord[1]. Failures in neurulation cause severe congenital malformations called neural tube defects (NTDs) which are amongst the most common human birth defects. In amniotes, the neural tube forms from head-to-tail, through two fundamentally different morphogenetic processes[1]. The rostral neural tube forms through folding and fusing of the neuroepithelium in a process called primary neurulation[2]. However, the caudal neural tube forms differently through secondary neurulation. This is characterised by condensation of cells to form a rod-like structure which undergoes canalisation to generate a lumen[3]. It is critical to nervous system function that these differing morphogenetic processes are integrated to form a continuous neural tube. The spatiotemporal dynamics of this process have not been determined due to a lack of live imaging and species differences in neurulation.

A junctional transition zone where primary and secondary neurulation appear to occur simultaneously across the dorsoventral axis, has been identified in avian[4,5] and human embryos[6,7]. More recently, a local morphogenetic process called junctional neurulation was proposed to occur in the avian junctional zone, ensuring continuity between the primary and secondary neural tubes[8]. Notably, most human spinal defects cluster near this junctional zone, highlighting its susceptibility to disruption[6]. In humans, impaired formation of the junctional neural tube leads to local spinal dysraphisms, recently named Junctional Neural Tube Defects (JNTDs)[9–12]. Patients with JNTDs exhibit structurally and functionally disconnected primary and secondary neural tubes, accompanied by lower limb deformities, impaired motor and sensory function and incontinence[9].

Mouse embryos lack a junctional zone[3,13], therefore our limited understanding of junctional neurulation has been derived from studies

[1]Institute for Molecular Bioscience, The University of Queensland, Brisbane, QLD, Australia. [2]Australian Institute for Bioengineering and Nanotechnology, The University of Queensland, Brisbane, QLD, Australia. [3]School of Chemistry and Molecular Biosciences, The University of Queensland, Brisbane, QLD, Australia. [4]School of Biomedical Sciences, The University of Queensland, Brisbane, QLD, Australia. ✉e-mail: melanie.white@imb.uq.edu.au

of avian embryos - primarily using fixed tissue sections or low resolution live imaging[8]. At the molecular level, the protein PRICKLE1 (PK1) has been implicated in junctional neurulation in avian embryos[8]. PK1 is a core protein in the Planar Cell Polarity (PCP) signalling pathway which controls convergent extension in vertebrates[14–16]. Planar-polarized enrichment of PCP proteins along mediolateral cell junctions drives actomyosin-dependent junction contraction and facilitates mediolateral cell convergence[17–20]. This PCP-dependent convergence of cells towards the embryonic midline is crucial to bend the neural plate and form the primary neural tube[18]. Disruption of other core PCP genes including *CELSR1-3*, *VANGL1/2* and *DVL2* can cause NTDs along the entire head-to-tail axis. *PRICKLE1* mutations, however, are predominantly associated with local spinal dysraphisms[21–25]. How PK1 disruption specifically affects the morphogenesis of the junctional neural tube remains unknown.

Here, we use high spatiotemporal resolution live imaging, targeted manipulations and quantitative cell tracking in transgenic quail embryos to characterise the real time cellular dynamics of junctional neurulation. We demonstrate that the junctional neural tube is formed by a combination of mediolateral cell convergence and epithelial-to-mesenchymal transition (EMT)-driven ingression of medial cells in the junctional zone. Disrupting PK1 impairs medial cell ingression and causes localized JNTDs by a novel PCP-independent mechanism that links EMT to neural tube closure. We find that PK1 localizes to the apical cortex of medial junctional zone cells, where it promotes actomyosin accumulation and apical constriction. Strikingly, PK1's control of cortical F-actin accumulation is uncoupled from planar cell polarity, but essential for both mediolateral convergence and EMT-driven ingression. Our findings reformulate the molecular logic of junctional neurulation and show how failure of EMT can cause localized defects in the junctional neural tube.

## Results

### Junctional neural tube closure requires PK1

To investigate the morphogenesis of the junctional neural tube, we used live imaging of a transgenic quail model[26,27]. The cellular material for the junctional zone between primary and secondary neurulation is generated in a region that is caudal and lateral to Hensen's node and rostral to the primitive streak[28,29], starting at the 5-somite stage (ss) in quail (Fig. 1a, b). Live imaging of Lifeact-EGFP transgenic quail embryos[26] reveals the junctional zone progressively converges from the 5ss until the 9-10ss when the posterior neuropore closes (Fig. 1b and Supplementary Movie 1). The cells of the junctional zone are SOX2[+] and give rise to both the junctional neural tube which forms between somites 18–27, many precursors for the secondary neural tube initiating from somite 28, a few cells in the caudal somites and cells in the tailbud (Fig. 1c, d and Supplementary Fig. 1a). To characterize the cellular dynamics driving junctional zone morphogenesis, we mosaically labelled cell nuclei by electroporating TgT2(CAG:NLS-mCherry-IRES-GFP-CAAX) quail embryos[27] with a plasmid encoding H2B-miRFP670. High-resolution live imaging combined with nuclear tracking revealed that cells converge from the lateral region of the junctional zone (defined as the outermost 100 μm at each lateral edge), towards the medial region consisting of 100 μm on each side of the midline (925 cell tracks in 3 embryos, Fig. 1e, f and Supplementary Movie 2). Within 3 h, the average convergent displacement was 16.26 ± 0.46 μm per cell, however cells in the lateral junctional zone showed significantly more convergent displacement (19.66 ± 1.84 μm) than those in the medial region (7.50 ± 0.96 μm, Fig. 1f). The cells in the lateral region also exhibited significantly stronger directionality towards the midline than those in the medial region (Fig. 1g). Further analysis revealed a gradient of mediolateral convergence (Fig. 1h). To examine PK1 expression, we performed immunofluorescent staining. Cells were computationally segmented using an N-cadherin signal and the PK1 intensity was measured per cell. Our analyses showed

expression of PK1 was significantly enriched in the junctional zone compared to the primary neural tube from the 5ss to the 9-10ss (Fig. 1i, j). To disrupt PK1 expression, we electroporated embryos with a DNA construct encoding an intronic miRNA targeting *PRICKLE1* (or a scrambled control miRNA). To quantify the knockdown, the PK1 miRNA (or scrambled control) was expressed upstream of a EGFP reporter[30] in -1ss wildtype quail embryos, which were immunostained for PK1 at the 6-7ss (Supplementary Fig. 1b–d). To examine the tissue-scale consequences of PK1 disruption, the PK1 miRNA (or scrambled control) was expressed upstream of H2B-miRFP670 in Lifeact-EGFP transgenic quails[26]. We followed the embryos by live imaging for -15 h, by which time the posterior neuropore of the control embryos was typically closed. Knocking down PK1 expression in the junctional zone disrupted junctional neurulation and prevented closure of the posterior neuropore (Fig. 1k, l, and Supplementary Fig. 1e, Supplementary Movie 3). We found comparable defects following electroporation of an siRNA targeting PK1 (Supplementary Fig. 1f–i). Together, these results confirm that correct formation of the junctional neural tube relies on PK1.

### PK1 knockdown disrupts cellular convergence without affecting planar cell polarity

To test how convergent cell movement within the junctional zone is impacted by PK1 knockdown, we electroporated -1ss TgT2(CAG:NLS-mCherry-IRES-GFP-CAAX) quail embryos with the plasmids encoding the PK1 miRNA or the scrambled control miRNA upstream of H2B-miRFP670. We performed live cell nuclei tracking from -7ss during junctional neurulation (Fig. 2a, b). PK1 knockdown cells showed significantly less convergent displacement (Fig. 2b–d, Supplementary Movie 4), and reduced directionality towards the midline (Fig. 2e) compared to scrambled control cells.

As PK1 is a core PCP protein, we determined whether the reduction in cell convergence following PK1 knockdown results from disrupted planar cell polarization. Immunofluorescent staining confirmed the PCP components VANGL2 and PK1 and their downstream effectors, ROCK1 and pMLC, and F-actin, are planar polarized, characterized by enrichment along mediolateral cell junctions in the junctional zone of the -5ss quail embryo (Supplementary Fig. 2a–c). However, unexpectedly, PK1 knockdown did not alter the planar polarity of its binding partner VANGL2, or the downstream effector ROCK1 or F-actin itself (Fig. 2f–h). Furthermore, PK1 knockdown in the junctional zone had no effect on the number or orientation of supracellular actin cables, which are regulated by PCP-dependent signalling in the avian neural tube (Fig. 2i–k)[18]. Together, these results suggest PK1 has a PCP-independent role in junctional neural tube formation.

### PK1 knockdown disrupts medial cell ingression

To understand how PK1 knockdown disrupts cell convergence without affecting planar cell polarization in the junctional zone, we examined cellular movements in the medial region, where cells typically converge to. By analysing the dorsoventral displacement of the H2B-miRFP670 labelled cell nuclei tracked in Fig. 1e, we found that cells within the medial region ingressed from the dorsal surface of the junctional zone into the underlying ventral tissue (Fig. 3a, b, Supplementary Fig. 3 and Supplementary Movie 5). Like the convergent displacement behavior, we observed a gradient of ventral displacement: within 3 h of live imaging, cells in the medial region moved ventrally (6.30 ± 0.70 μm), while cells in the lateral region moved dorsally (3.49 ± 1.02 μm, Fig. 3c, d). The medial cells also exhibited significantly stronger ventral directionality than cells in the lateral regions (Fig. 3e). Live imaging of the dorsal surface of the junctional zone in Lifeact-EGFP quail embryos at -5ss enabled visualization of the apical cell surfaces (Fig. 3f). Cells in the medial region accumulated cortical LifeAct-EGFP and reduced their apical surface area over time before ingressing and disappearing from the dorsal surface (Fig. 3f, g).

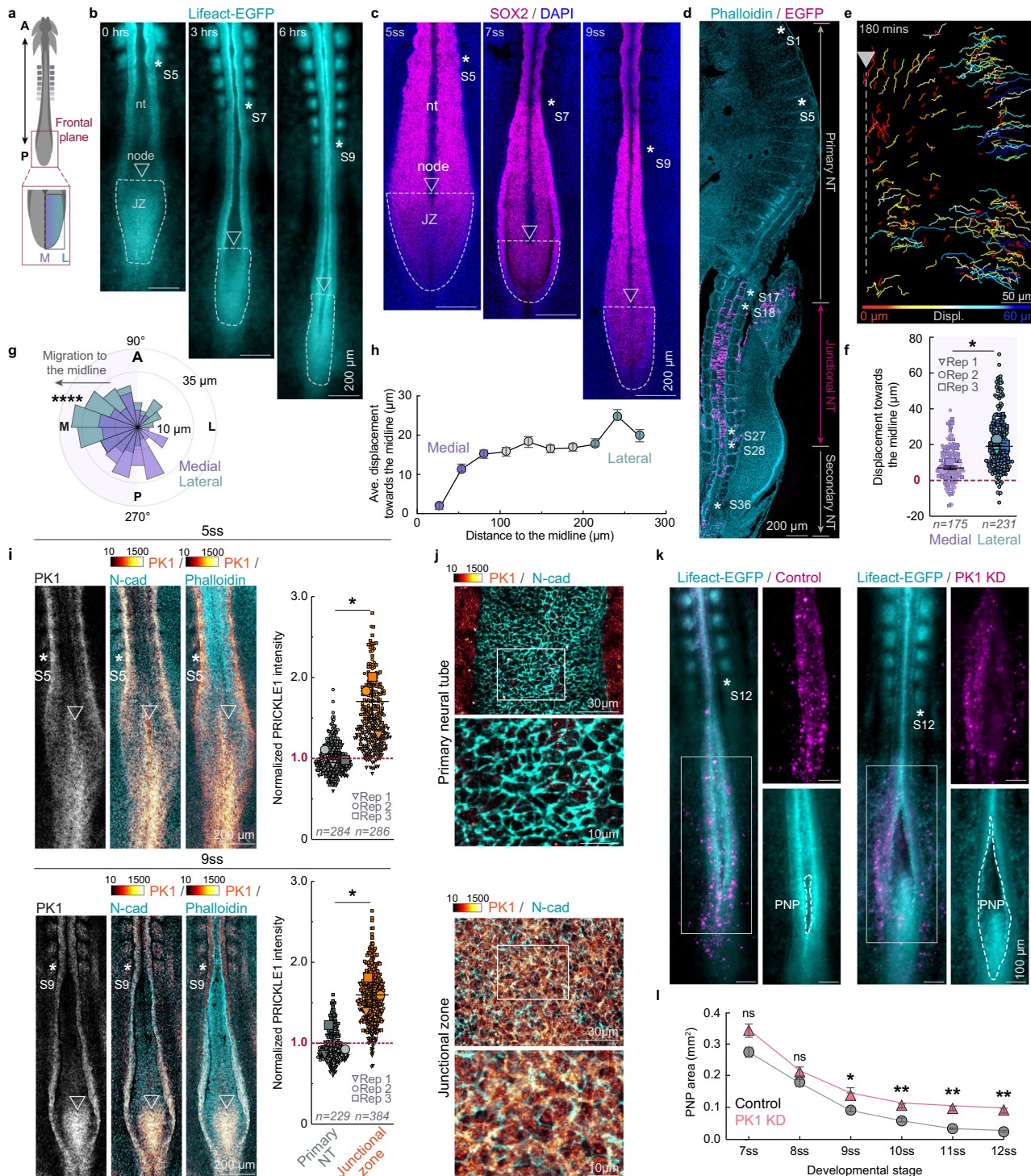

**Fig. 1 | Junctional neural tube closure is PK1-dependent. a** Schematic showing region of interest. A: anterior, P: posterior, M: medial, L: lateral. **b** Location of junctional zone (JZ) from the 5ss to the 9ss in a Lifeact-EGFP embryo. **c** Location of JZ in fixed wildtype embryos stained with SOX2 and DAPI. **d** Wildtype embryo electroporated with EGFP mRNA in JZ and incubated for 48 h. JZ cells contribute to junctional NT (somite 17/18 – 27/28) and secondary NT (> somite 27/28). **e** Snapshot showing individual tracks of migrating H2B-miRFP670 labelled cells. **f** Displacement towards the midline of cells in the medial ($n = 175$ cells from 3 embryos) and lateral regions ($n = 231$ cells from 3 embryos) mean, ± sem, two-sided paired $t$-test, $p = 0.0331$. **g** Directionality of cell trajectories in the medial ($n = 175$ cells from 3 embryos) and lateral ($n = 231$ cells from 3 embryos) regions, $p = 8.3 \times 10^{-33}$ two-sided Kolmorogov–Sminorv test. **h** Cell displacement towards

the midline plotted against cell starting positions across the mediolateral axis ($n = 925$ tracked cells from 3 embryos) mean, ± sem. **i** PK1 is highly expressed in JZ compared to primary NT, mean, ± sem, (for 5ss, $n = 284$ cells from 3 embryos in primary NT and $n = 286$ cells from 3 embryos in junctional zone), $p = 0.0389$ (for 9ss, $n = 229$ cells from 3 embryos in primary NT and $n = 384$ cells from 3 embryos in junctional zone), $p = 0.0239$, two-sided paired $t$-test. **j** Immunolabelling of the primary NT and JZ with N-cadherin and PK1. **k, l** PK1 knockdown impairs posterior neuropore (PNP) closure, mean, ± sem, ($n = 3$ embryos for both control and KD), $p = 0.0932, 0.3359, 0.0174, 0.0094, 0.0017, 0.0024$, for 7ss to 12ss, respectively, two-sided un-paired $t$-test. nt: neural tube. Asterisks indicate somite positions. Triangles indicate node position. Color scales depict pixel intensity [a.u.]. Displ: displacement.

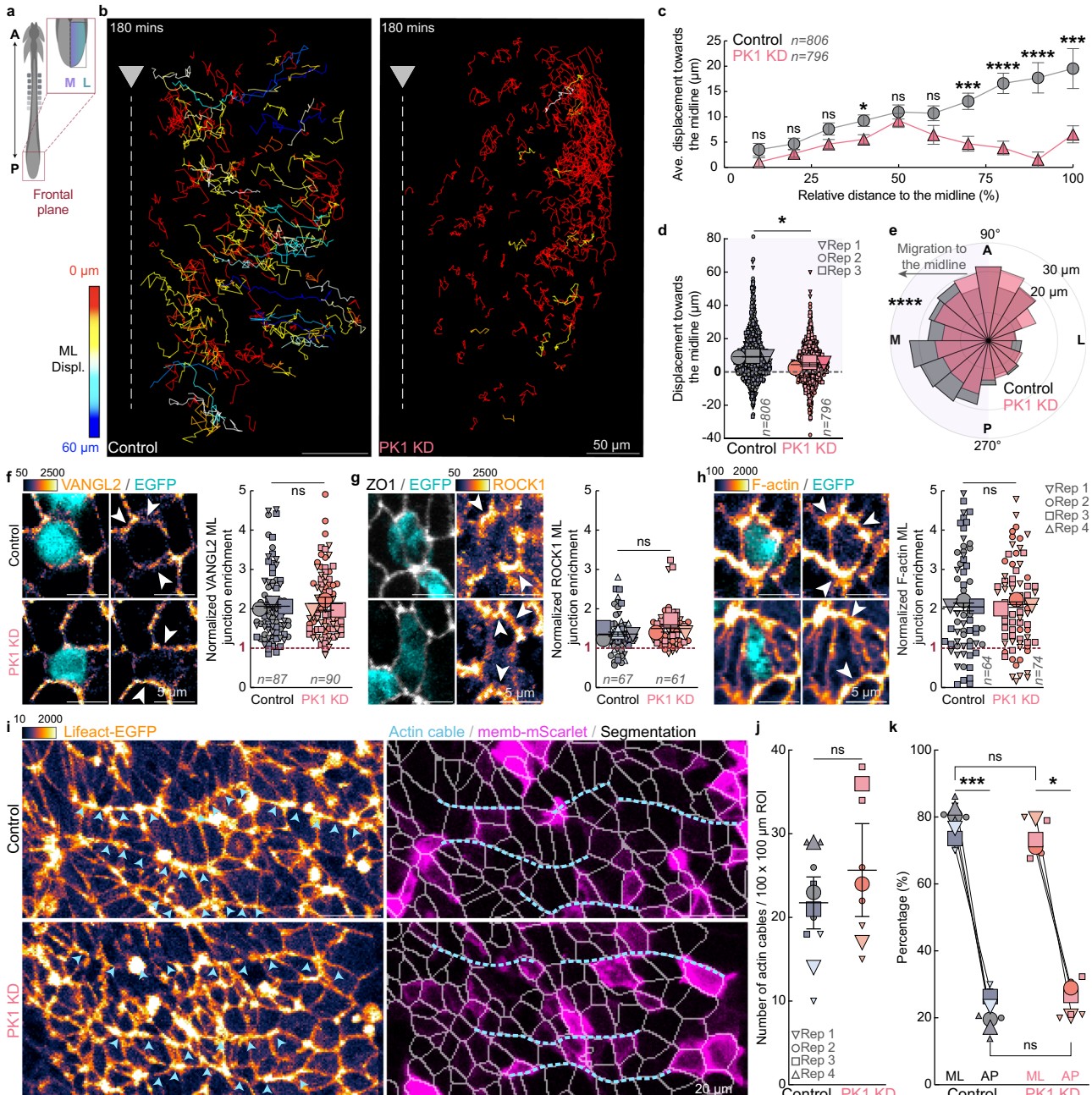

**Fig. 2 | PK1 knockdown disrupts cellular convergence without affecting planar cell polarity. a** Schematic shows region of interest. A: anterior, P: posterior, M: medial, L: lateral. **b** Live imaging showing cell trajectories color-coded by ML displacement in Control and PK1 KD embryos. **c** Cell displacement towards the midline plotted against cell starting positions across the mediolateral axis for PK1 knockdown (796 tracked cells from 3 embryos) and Control cells (n = 806 cells from 3 embryos, mean, ± sem), p = 0.1109, 0.1479, 0.0536, 0.0182, 0.4161, 0.0723, 0.0003, $1.9 \times 10^{-7}$, $3.8 \times 10^{-7}$, 0.0008 for 10% to 100% distance to the ML, two-sided un-paired t-test. **d** Displacement towards the midline of PK1 knockdown (796 tracked cells from 3 embryos) and Control cells (n = 806 cells from 3 embryos), mean, ± sem, p = 0.0137, two-sided un-paired t-test. **e** Directionality of PK1 knockdown (796 tracked cells from 3 embryos) and Control cells (n = 806 cells from 3 embryos), p = $3.8 \times 10^{-19}$, two-sided Kolmorogov–Sminorv test. **f–h** Immunofluorescent

staining of PCP protein VANGL2 and downstream effectors ROCK1 and F-actin reveals no change in planar polarisation following PK1 knockdown. mean, ± sem, p = 0.7255, 0.2030, 0.9765 for **f–h**, ns, not significant, two-sided un-paired t-test. ML; mediolateral **i** Segmentation and identification of mediolaterally oriented supracellular actin cables in knockdown or control embryos in the JZ. **j** PK1 knockdown does not alter number of actin cables, mean, ± sem, (n = 4 embryos for control and 3 embryos for PK1 KD, 2 ROIs per embryo), p = 0.5369, ns, not significant, two-sided un-paired t-test. **k** PK1 knockdown does not alter the orientation of actin cables, mean, ± sem, (n = 4 embryos for control and 3 embryos for PK1 KD, 2 ROIs per embryo), p = 0.0008 and 0.0105 for ML vs AP in Control and PK1 KD, respectively, two-sided paired t-test. p = 0.5369 and 0.5369 for Control vs PK1 KD of ML and AP, respectively, ns, not significant, two-sided un-paired t-test. Triangles indicate node position. Color scales depict pixel intensity [a.u]. Displ: displacement.

Significantly more cells ingressed in the medial region than the lateral regions (Fig. 3h).

To determine if PK1 knockdown affects the medial cell ingression, we electroporated ~1ss embryos with the PK1 miRNA-H2B-miRFP670 construct (or the scrambled control) and tracked cell nuclei by live

imaging from ~7ss. PK1 knockdown significantly reduced cell ingression in the medial region compared to scrambled controls (Fig. 3i–k and Supplementary Movie 6). This was accompanied by a significant loss of ventral directionality (Fig. 3l), confirming that PK1 is required for cell ingression in the medial region of the junctional zone.

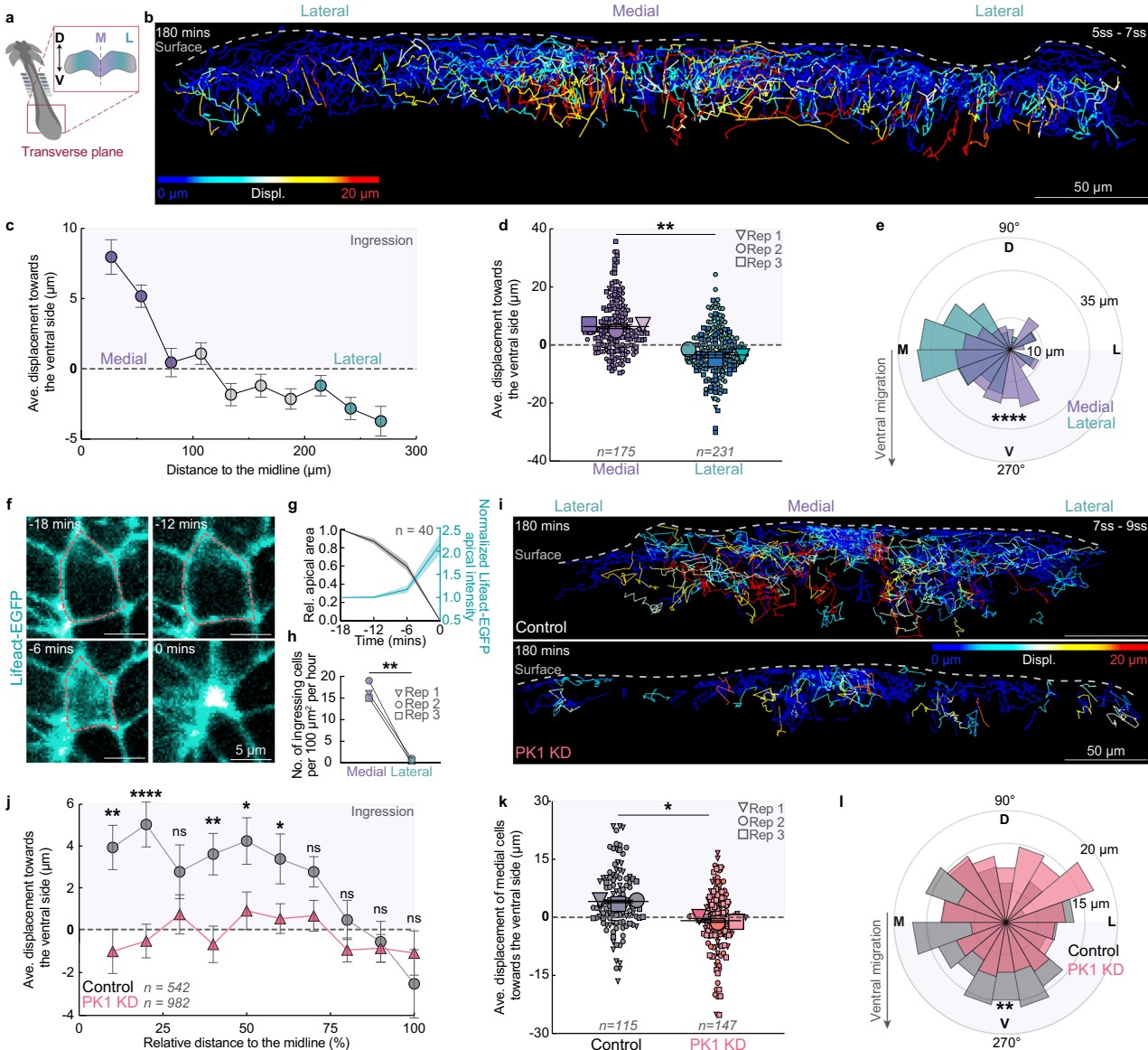

**Fig. 3 | PK1 knockdown disrupts medial cell ingression. a** Schematic showing region of interest. D: dorsal, V: ventral, M: medial, L: lateral. **b** Snapshot showing individual tracks of H2B-miRFP670 labelled cells migrating along the dorsoventral axis. **c** Ventral cell displacement plotted against cell starting positions across the mediolateral axis ($n = 927$ tracked cells from 3 embryos), mean, ± sem. **d** Ventral displacement of cells in the medial ($n = 175$ cells from 3 embryos) and lateral ($n = 231$ cells from 3 embryos) regions. mean, ± sem, $p = 0.0023$, two-sided paired $t$-test. **e** Directionality of cell trajectories along the dorsoventral (D, V) and mediolateral (M, L) axes in the medial ($n = 175$ cells from 3 embryos) and lateral (n = 231 cells from 3 embryos) regions, $p = 3.9 \times 10^{-18}$, two-sided Kolmorogov–Sminorv test. **f** Accumulation of Lifeact-EGFP at the shrinking apical cortex of an ingressing cell. **g** Lifeact-EGFP accumulates at the apical cortex as the apical cell area shrinks during cell ingression, mean, ± sem ($n = 40$ cells from 4 embryos). **h** Significantly more

cells ingress in the medial JZ region than the lateral, ($n = 3$ embryos), $p = 0.0066$, two-sided paired $t$-test. **i** Snapshots showing dorsoventral trajectories of individual Scrambled Control miRNA-H2B-miRFP670 or PK1 miRNA-H2B-miRFP670 labelled cells. **j** Ventral displacement plotted against cell starting positions across the mediolateral axis for PK1 knockdown (982 tracked cells from 3 embryos) and Control cells ($n = 542$ cells from 3 embryos), mean, ± sem, $p = 0.0012$, $4.4 \times 10^{-5}$, 0.1983, 0.0015, 0.0198, 0.0315, 0.0555, 0.2048, 0.8301, 0.4726 for 10% to 100% distance to the midline, two-sided un-paired $t$-test. **k** Ventral displacement of PK1 knockdown ($n = 115$ cells from 3 embryos) and Control cells ($n = 147$ cells from 3 embryos), mean, ± sem, $p = 0.0011$, two-sided un-paired $t$-test. **l** Directionality of PK1 knockdown ($n = 115$ cells from 3 embryos) and Control cells ($n = 147$ cells from 3 embryos) along the dorsoventral (D, V) and mediolateral (M, L) axes, p = 0.0011, two-sided Kolmorogov–Sminorv test. Displ: displacement.

## Medial cells undergo an epithelial-to-mesenchyme transition

PK1 has not been previously linked to cell ingression, therefore, we conducted a more detailed investigation into the mechanisms causing medial cells to ingress. The junctional zone is situated immediately rostral to the primitive streak: a region of ongoing cell ingression. Cells in the primitive streak undergo an epithelial-to-mesenchymal transition (EMT) during which they constrict their apical surfaces, ingress into the deeper tissue layer and switch from expression of the epithelial cadherin E-cadherin to the mesenchymal cadherin,

N-cadherin[31,32]. Therefore, we hypothesized that a similar mechanism may drive the ingression of cells in the medial region of the junctional zone.

The *SNAIL* gene family member, SLUG, is essential for EMT and cell ingression at the primitive streak[33]. Immunofluorescent staining revealed that cells in the junctional zone already express N-cadherin and not E-cadherin (Supplementary Fig. 4a), but many cells localized to the medial region were SLUG+ (Fig. 4a). To confirm that the SLUG+ cells in the medial region ingress, we used a SLUG enhancer construct that

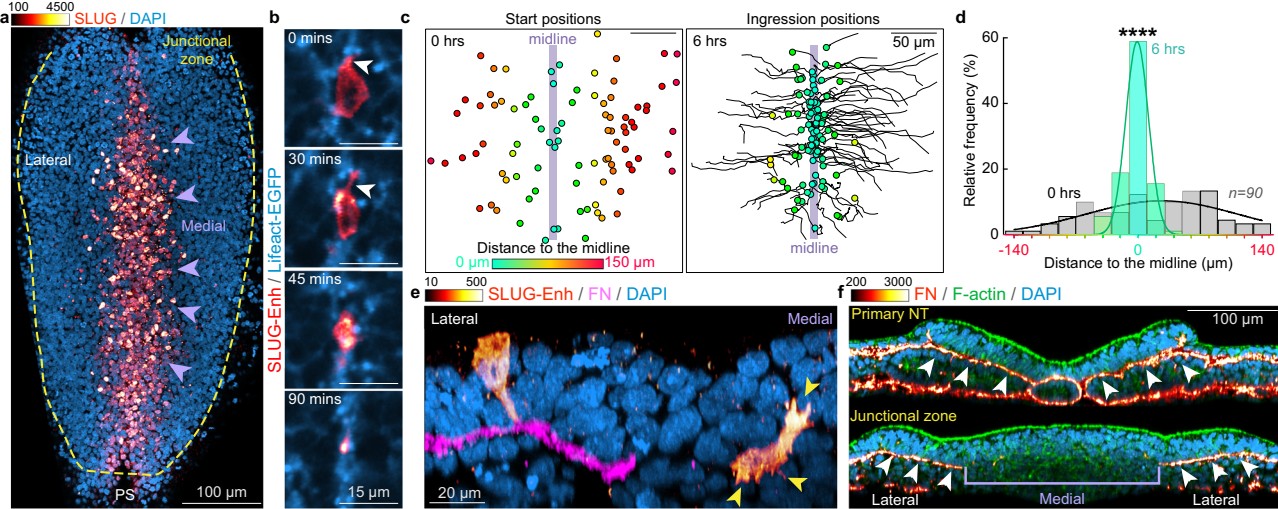

**Fig. 4 | Medial junctional zone cells undergo EMT and ingress.**
**a** Immunofluorescent staining showing SLUG expression in medial JZ cells
(arrowheads). PS; primitive streak. **b** Live imaging of a medial cell expressing a
SLUG reporter constricting its apical surface, extending protrusions (arrowheads)
and ingressing. **c, d** Positions of SLUG enhancer-positive cells at the onset of live
imaging (0 h, left in c) and their ingression sites near the midline (6 h, right in c).
SLUG enhancer-positive cells are initially distributed across the medial region of the
JZ (grey bars), but ingress near the midline (green bars), ($n = 90$ cells from 4
embryos), $p = 2.4 \times 10^{-9}$, two-sided Kolmorogov–Sminorv test. **e** SLUG enhancer-
positive cells have an epithelial morphology in the lateral JZ but highly protrusive
(arrowheads) mesenchymal morphology in the medial JZ. **f** Immunofluorescent
staining reveals a continuous layer of FN underneath the primary neural plate (top,
arrowheads) which is absent in the medial region of the JZ (bottom, bracket). Color
scales depict intensity [a.u.].

reports SLUG expression[34] (Supplementary Fig. 4b). We electro-
porated 1ss Lifeact-EGFP quail embryos with a DNA plasmid that
expresses membrane-mScarlet under the control of the SLUG enhan-
cer (SLUG enh-memb-mScarlet) and followed the cells by live imaging
from ~5ss. Many cells within the junctional zone were labelled with
SLUG enh-memb-mScarlet and visibly reduced their apical surface area
over time before disappearing from the dorsal side (Fig. 4b). The SLUG
enhancer-positive cells were initially distributed across the junctional
zone but within 6 h, moved to within 50 μm of the midline, where they
ingressed (Fig. 4c, d and Supplementary Movie 7). This medial move-
ment was significantly faster than cells labelled with a ubiquitous CMV
promoter (Supplementary Fig. 4c). Furthermore, there was a mor-
phological change from an epithelial profile in the lateral region to a
highly protrusive mesenchymal morphology in the medial region
(Fig. 4e and Supplementary Fig. 4d).

Another crucial early step in EMT in the primitive streak is
degradation of the basement membrane[31]. During primary neurula-
tion, the ventral side of the neural plate is covered by a layer of the
extracellular matrix proteins Fibronectin (FN) and Laminin (Fig. 4f and
Supplementary Fig. 4e, top). This layer remains intact in the lateral
regions of the junctional zone but is absent in the medial region,
consistent with EMT-driven medial cell ingression (Fig. 4f and Sup-
plementary Fig. 4e, bottom). Together, these findings demonstrate
that cells in the junctional zone of the neural plate undergo at least a
partial EMT once they reach the medial region to ingress towards the
ventral side.

### Medial cell ingression is required for junctional neural tube formation

We next asked whether the medial cell ingression is necessary for the
correct formation of the junctional neural tube. To block EMT and
medial cell ingression, we co-electroporated a morpholino targeting
SLUG (or a non-targeting control morpholino) with a plasmid encod-
ing H2B-EGFP into the junctional zone at the 1ss. We then fixed and
stained the embryos for SLUG and DAPI during junctional neurulation
at the 7ss (Supplementary Fig. 5a). In control embryos, 37.23 ± 7.22 %
(mean ± sem) of the electroporated cells in the medial junctional zone

were SLUG$^+$. This was reduced to 8.56 ± 1.29 % of electroporated medial
cells in the SLUG knockdown embryos (Fig. 5a). Additionally, SLUG
knockdown significantly reduced the proportion of cells with a pro-
trusive mesenchymal morphology (Fig. 5b, c). Together these results
confirm the efficacy of the morpholino. In control embryos, the H2B-
EGFP-positive cells were distributed across the dorsoventral axis of the
junctional zone and 23.6 ± 6.0 % of cells ingressed to the ventral third
of the tissue (Fig. 5d, e). However, in SLUG knockdown embryos, the
electroporated cells remained significantly closer to the dorsal surface
and only 11.9 ± 5.0 % of H2B-GFP-positive cells penetrated to the ven-
tral third of the junctional zone (Fig. 5d, e), confirming that SLUG
expression is required for medial cell ingression.

Fibroblast Growth Factor (FGF) signalling is required for the
expression of SLUG and cell ingression at the primitive streak in mouse
and chick embryos[35,36]. To determine if FGF also drives SLUG expres-
sion in the junctional zone, we treated 1ss wildtype embryos with
10 μM of the FGF Receptor inhibitor Infigratinib[37] or a DMSO control
and then electroporated the junctional zone with the H2B-EGFP plas-
mid to assay for cell ingression. To confirm that blocking FGF signal-
ling reduced SLUG expression, we fixed embryos at the 5ss when
junctional neurulation initiates and stained for SLUG. In the control
embryos, 74.70 ± 3.09 % (mean ± sem) of cells in the medial junctional
zone were SLUG$^+$, whereas this reduced to 51.51 ± 4.93 % in the FGF-
inhibited embryos (Fig. 5f, g). Consistent with the effects of reducing
SLUG expression, in the FGF-inhibited embryos the H2B-EGFP labelled
cells remained significantly closer to the dorsal surface of the junc-
tional zone than in the DMSO control embryos and no cells penetrated
to the ventral third of the tissue (Fig. 5h, i). Furthermore, by the 9-10ss
the posterior neuropore had typically closed in control embryos with
an average width of 45.40 ± 17.27 μm. By contrast, the posterior neu-
ropore remained open in FGF-inhibited embryos (average width of
183.50 ± 8.91 μm), resulting in a junctional neural tube defect (Fig. 5j,
k). To exclude potential confounding effects of FGF inhibition in sur-
rounding tissues, we electroporated plasmids encoding a dominant-
negative FGFR1[38] into the neural tube of 1ss wildtype quail embryos.
This more targeted inhibition also reduced the number of SLUG$^+$ cells
in the junctional zone, inhibited the ingression of medial cells and

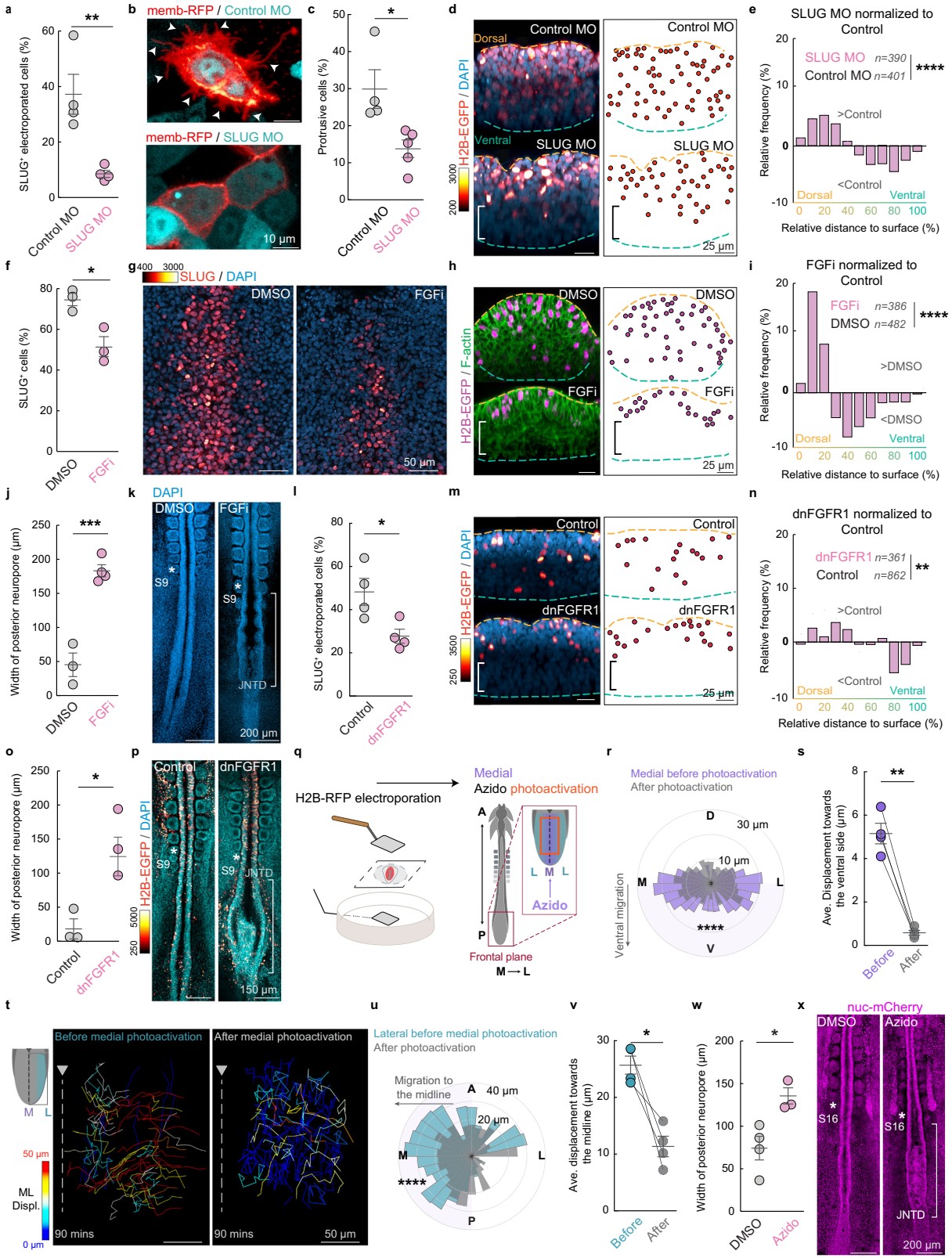

resulted in junctional neural tube defects characterised by an open posterior neuropore (Fig. 5l–p and Supplementary Fig. 5b). These findings confirm that FGF-dependent SLUG expression is required in the junctional zone to drive ingression of the medial cells and enable posterior neuropore closure. Blocking FGF signalling specifically impacted junctional neurulation and did not disrupt the neural tube before the 6ss (Fig. 5k, p) or when the inhibitor was added after PNP

closure (Supplementary Fig. 5c). Furthermore, FGF inhibition did not block the assembly of actin cables associated with convergence in the lateral regions of the junctional zone (Supplementary Fig. 5d), suggesting the defect primarily results from the prevention of medial cell ingression in the junctional zone.

Prior to ingression, the medial cells undergo apical constriction (Fig. 3f, g), a process that depends on actomyosin contraction[39].

**Fig. 5 | Medial cell ingression is required for junctional neural tube formation.**
**a** Electroporation of a SLUG morpholino (MO) effectively reduces the percentage of SLUG⁺ cells in the JZ, mean, ± sem, (*n* = 4 embryos for both Control and SLUG MO), *p* = 0.0079, two-sided un-paired *t*-test. **b**, **c** Electroporation of a SLUG MO reduces the protrusive morphology of cells in the JZ, mean, ± sem, (*n* = 4 embryos for Control and *n* = 5 embryos for SLUG MO), *p* = 0.0188, two-sided un-paired *t*-test. **d** SLUG knockdown reduces ingression of electroporated cells to the ventral side (bracket). **e** Relative frequencies of SLUG knockdown cells normalized to control cells along the dorsoventral axis. SLUG knockdown cells (390 cells from 4 embryos) are significantly more frequently located near the dorsal surface than Control cells (401 cells from 4 embryos), *p* = 2.2 × 10⁻⁷, two-sided Kolmorogov–Sminorv test. **f**, **g** FGF inhibition significantly reduces the percentage of SLUG⁺ cells in the JZ, mean, ± sem, (*n* = 3 embryos for both DMSO and FGFi), *p* = 0.0163, two-sided un-paired *t*-test. **h** FGF inhibition prevents ingression of electroporated medial JZ cells (magenta) to the ventral side of the tissue (bracket). **i** Relative frequencies of cells in embryos treated with FGF inhibition normalized to DMSO control along the dorsoventral axis. Cells in FGF inhibitor-treated embryos (*n* = 386 cells from 3 embryos) are significantly more frequently located near the dorsal surface than DMSO control-treated cells (*n* = 482 cells from 3 embryos), *p* = 3.5 × 10⁻¹⁷, two-sided Kolmorogov–Sminorv test. **j** The posterior neuropore is significantly wider in 9 – 10 somite stage FGF inhibitor-treated embryos than the DMSO-treated controls, mean, ± sem, (n = 3 embryos for DMSO and 4 embryos for FGFi), *p* = 0.006, two-sided un-paired t-test. **k** FGF inhibition causes a junctional neural tube defect (JNTD, white bracket). **l** Electroporation of a dnFGFR1 construct effectively reduces the percentage of SLUG⁺ cells in the JZ, mean, ± sem, (*n* = 4 embryos for both Control

and dnFGFR1), *p* = 0.0266, two-sided un-paired *t*-test. **m** The dnFGFR1 reduces ingression of electroporated cells to the ventral side (bracket). **n** Relative frequencies of dnFGFR1 cells normalized to control cells along the dorsoventral axis. dnFGFR1 cells (*n* = 361 cells from 3 embryos) are significantly more frequently located near the dorsal surface than control cells (*n* = 862 cells from 3 embryos). *p* = 0.0011, two-sided Kolmorogov–Sminorv test. **o** The posterior neuropore is significantly wider in 9 – 10 somite stage dnFGFR1 expressing embryos than the controls, mean, ± sem, (*n* = 3 embryos for both Control and dnFGFR1), *p* = 0.0179, two-sided un-paired *t*-test. **p** dnFGFR1 causes a junctional neural tube defect (JNTD, white bracket). **q** Schematic of experimental approach to inhibit myosin II by photoactivation of azidoblebbistatin in the JZ and track cell movements. A: anterior, P: posterior, M: medial, L: lateral. **r**, **s** Photoactivating azidoblebbistatin in the medial JZ decreases medial cell ingression (*n* = 403 cells from 4 embryos in before activation condition and *n* = 405 cells from 4 embryos in after activation condition), for r, *p* = 3.8 × 10⁻¹⁴, two-sided Kolmorogov–Sminorv test; for s, mean, ± sem, p = 0.0013, two-sided paired *t*-test. **t**–**v** Photoactivating azidoblebbistatin in the medial JZ reduces lateral cell convergence to the midline (*n* = 500 cells from 4 embryos in before activation condition and *n* = 433 cells from 4 embryos for after activation condition), for u, p = 1.17 × 10⁻⁵, two-sided Kolmorogov–Sminorv test; for v, mean, ± sem, *p* = 0.0183, two-sided paired *t*-test. **w**, **x** Photoactivating azidoblebbistatin in the medial JZ results in a significantly wider posterior neuropore compared to DMSO-treated controls and causes a JNTD (white bracket), mean, ± sem, (*n* = 4 embryos for DMSO and *n* = 3 embryos for Azido), *p* = 0.0195, two-sided un-paired *t*-test. Color scales depict pixel intensity [a.u.]. Displ: displacement.

Therefore, to selectively target cell ingression in the medial region of the junctional zone, we used a photo-activatable myosin II inhibitor, azidoblebbistatin[40,41]. First, we electroporated 1ss Lifeact-EGFP quail embryos with a plasmid encoding H2B-RFP into the junctional zone to facilitate cell tracking. Next, we applied azidoblebbistatin at the 5ss and specifically activated it in the medial region using a 405 nm laser (Fig. 5q). Live cell tracking revealed a significant decrease in the ventral displacement of medial cells following photoactivation (from 5.16 ± 0.47 µm to 0.58 ± 0.11 µm), confirming that blocking actomyosin contraction inhibited medial cell ingression (Fig. 5r, s). Notably, although myosin II inhibition was restricted to the medial region, lateral cells also exhibited a significant reduction in convergence toward the midline (Fig. 5t–v, from 25.71 ± 1.59 µm to 11.39 ± 1.82 µm). These findings suggest that the ingression of medial cells is required to create space that allows subsequent convergence of lateral cells into the midline. To examine the effect at the tissue scale, we treated TgT2(CAG:NLS-mCherry-IRES-GFP-CAAX) embryos at the 5ss with azidoblebbistatin. Consistent with the EMT inhibition, blocking actomyosin contraction in only the medial cells caused focalised junctional NTDs, characterised by a significantly wider posterior neuropore at the ~16ss (Fig. 5w, x). Thus, medial cell ingression is required for the correct formation of the junctional neural tube.

## PK1 regulates cortical actomyosin in a PCP-independent manner for medial cell ingression

To understand PK1's role in medial cell ingression we first tested whether PK1 controls SLUG expression. We electroporated 1ss wild-type quail embryos with the plasmids encoding the intronic PK1 miRNA or the scrambled control miRNA upstream of EGFP. We fixed and immunostained the embryos for SLUG at 5ss and found that knocking down PK1 did not change SLUG expression levels or the number of SLUG⁺ cells (Supplementary Fig. 6a–c). This confirms that PK1 does not regulate SLUG expression in the junctional zone.

To investigate how PK1 knockdown affects the cell apical surface area, we expressed the PK1 miRNA or the scrambled control miRNA upstream of membrane-mScarlet in the junctional zone of Lifeact-EGFP quail embryos and examined the distribution of apical cell surface size at the 5ss. We used a custom local Z projection strategy to project the curved dorsal surface of the junctional zone onto a 2D plane and computationally segmented cell boundaries labelled by

Lifeact-EGFP[26]. Quantification revealed the average apical surface area of PK1 knockdown cells was significantly larger than scrambled control cells (Fig. 6a, b). To determine if the dilated apical cell area results from defects in apical constriction, individual cells were tracked for ~3 h and the change in apical surface area over time was analysed[42]. A closer examination of the medial cell apical area dynamics revealed no differences between control cells and PK1 knockdown cells during the non-constriction phase. However, during the constriction phase, PK1 knockdown cells displayed a significantly smaller reduction in apical surface area compared to controls (Fig. 6c–e and Supplementary Movie 8). In the control embryos, all tracked medial cells shrunk their apical surfaces within 3 h and 40% of them fully ingressed (Fig. 6c, d). By contrast, in the PK1 knockdown embryos, 90% of the tracked medial cells reduced their apical surface by less than 35%, and none shrank by more than 55% (Fig. 6c–e). Furthermore, none of the PK1 knockdown cells ingressed within the 3 h observation period. These findings demonstrate that PK1 knockdown specifically inhibits medial cell apical constriction, preventing ingression.

Apical constriction is associated with enrichment of actomyosin at the apical cortex[39] and PK1 has recently been shown to regulate cortical F-actin in *Xenopus* embryos[43]. To characterise PK1 at the apical cortex in the junctional zone, we fixed wildtype quail embryos at the 5ss and immunostained for activated Myosin Light Chain (p-MLC), PK1 and N-cad to label the cell junctions (Fig. 7a, b and Supplementary Fig 7a, b). We used the Fiji plugin LocalZProjector[44] to generate a surface-restricted 2D projection to selectively visualise the apical ~3 µm of the tissue, using the N-cad signal to define the apical plane[42]. Large patches of p-MLC and F-actin were present at the apical cortex of 22–69% of medial cells (Fig. 7c). Although the proportion of medial cells with apical actomyosin patches varied among embryos, nonetheless, there were always significantly fewer cells with enriched apical actomyosin in the lateral regions of the junctional zone (2.5–10.5% of lateral cells, 4 embryos, Fig. 7c). In *Xenopus* mesodermal cells, it is the diffuse cytoplasmic pool of PK1 that increases cortical F-actin[43]. Therefore, we examined PK1 subcellular localisation near the cortex of cells in the junctional zone. Immunofluorescent staining showed significantly higher levels of PK1 at the apical cortex of medial cells compared to lateral cells in the junctional zone (Fig. 7d–g). Enrichment of apical F-actin was often observed in the medial cells with high cortical PK1 (Fig. 7f). To understand how PK1 affects the cortical

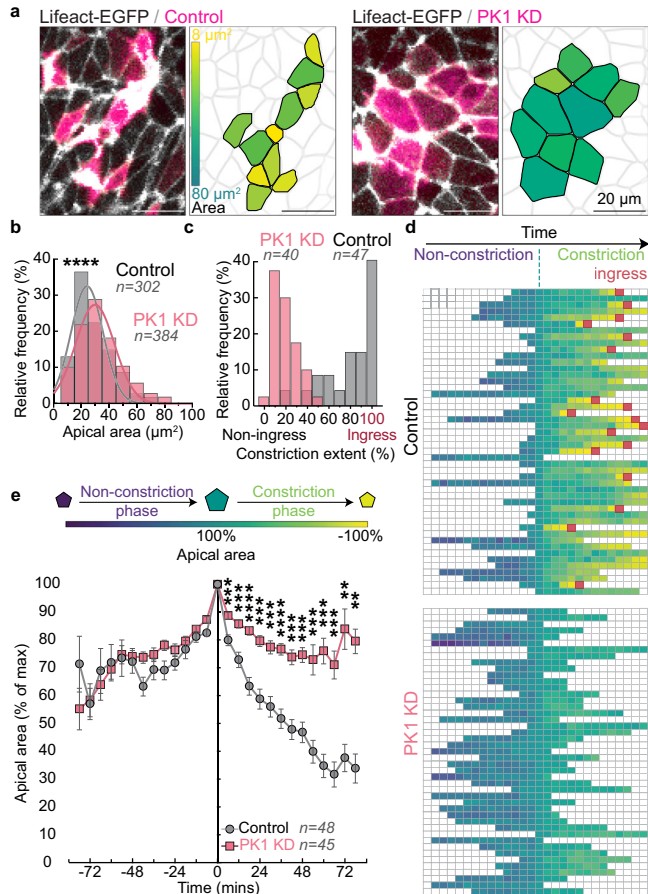

**Fig. 6 | Ingression of medial junctional zone cells requires PK1-dependent apical constriction. a, b** PK1 knockdown causes a significant increase in JZ cell apical area. ($n$ = 384 cells from 3 PK1 knockdown embryos and $n$ = 302 cells from 3 control embryos), $p$ = 1.2 × 10$^{-4}$, two-sided Kolmorogov–Sminorv test. **c** PK1 KD cells ($n$ = 40 cells from 3 embryos) constrict their apical surface less than Control cells ($n$ = 47 cells from 3 embryos) and fail to ingress. **d** Individual Control and PK1 KD cells colour-coded by apical surface area over time, aligned by their maximal apical area. The red squares indicate ingression events. **e** Quantification of cell apical area during live imaging. Individual cells are aligned to t = 0 min when they reach their maximal area. The negative time refers to the preceding non-constriction phase, and the positive time represents the constriction phase when the cell apical surface shrinks ($n$ = 45 cells from 3 PK1 knockdown embryos and 48 cells from 3 control embryos), mean, ± sem, $p$ = 0.0004, 0.0003, 4.88 × 10$^{-6}$, 0.0001, 0.0002, 7.17 × 10$^{-5}$, 0.0002, 0.0001, 0.0005, 0.0001, 0.0006, 0.0033, 0.0040 for 6 min to 78 min, two-sided un-paired $t$-test.

accumulation of actin in the quail embryo in real time, we electroporated the junctional zone of Lifeact-EGFP embryos with the PK1 miRNA or the scrambled control miRNA upstream of membrane-mScarlet at the 1ss and performed live imaging from 5ss. In the scrambled control cells, there was a significant increase in Lifeact-EGFP intensity at the apical cortex, concomitant with a significant reduction in apical surface area (Fig. 7h, i and Supplementary Fig. 7c). The PK1 knockdown cells, however, did not show any change in Lifeact-EGFP intensity or apical surface area, confirming that PK1 is required for cortical accumulation of Lifeact-EGFP and apical constriction.

Our findings reveal that correct formation of the junctional neural tube requires (1) ingression of SLUG⁺ medial cells and (2) PK1-dependent cortical actin accumulation and apical constriction. However, PK1 does not alter SLUG expression. This led us to finally ask if SLUG⁺ cells rely on PK1-mediated apical constriction to ingress. To address this question, we electroporated wildtype embryos with the

PK1 miRNA or the scrambled control miRNA upstream of H2B-miRFP670 at the 1ss. We then fixed and immunostained for SLUG at the 7ss to identify ingressing cells. Quantitative analysis demonstrated that whereas SLUG⁺/Control cells were distributed throughout the dorsoventral axis of the junctional zone, SLUG⁺/PK1 KD cells remained significantly closer to the dorsal surface and did not penetrate at all to the most ventral tissue (Fig. 7j, k). These results demonstrate that SLUG⁺ medial cells rely on PK1-driven apical constriction to successfully ingress. Disruption of PK1 impairs SLUG⁺ cell apical constriction and ingression, ultimately leading to junctional neural tube defects.

## Discussion

Junctional neural tube closure is a critical morphogenetic event bridging primary and secondary neurulation. By combining high resolution live imaging, targeted manipulations, and quantitative cell tracking in transgenic quail embryos, we reveal that PK1 is necessary for both robust mediolateral cell convergence and for the apical constriction–dependent ingression of medial junctional zone cells. Importantly, these functions of PK1 are executed independently of classical Planar Cell Polarization. Therefore, we have established a novel, PCP-independent role for PK1 in neural tube morphogenesis. Given that impaired junctional neural tube formation underlies human junctional neural tube defects[9–12], and that *PRICKLE1* mutations have been linked to local spinal dysraphisms[21], our findings provide a mechanistic basis for how PK1 dysfunction may contribute to human disease.

### Medial cell ingression couples EMT to neural tube closure

Our live imaging and cell tracking reveal that junctional zone cells initially converge mediolaterally before undergoing dorsoventral ingression into the underlying ventral tissue (Figs. 2, 3). Medial cells express the EMT regulator SLUG, constrict their apical surfaces and adopt protrusive, motile morphologies reminiscent of cells ingressing at the primitive streak (Fig. 4). Disruption of SLUG function, either by morpholino knockdown or FGF-inhibition, prevents medial cell ingression and yields focal junctional neural tube defects (Fig. 5), demonstrating that EMT-like behaviour is essential for neural tube closure. Traditionally, EMT-mediated ingression at the primitive streak is viewed as a hallmark of gastrulation, distinguishing mesendodermal-fated cells from adjacent non-ingressing neuroectoderm[45]. However, in the junctional zone we found that medial neuroectoderm cells also undergo a similar EMT and ingress. Yet, these cells largely retain neural identity and ultimately contribute to both the junctional and secondary neural tubes with only very minor contribution to the mesoderm (Fig. 1d). This uncoupling of EMT from lineage transition suggests that, unlike in the primitive streak, EMT within the junctional zone primarily serves a morphogenetic function, facilitating tissue remodelling and neural tube elongation, rather than as a mechanism to effect a change in cell fate.

### A novel, PCP-independent role for PK1

PK1 is traditionally classified as a core component of the PCP pathway, where it contributes to convergent extension movements in multiple vertebrate systems[46–49]. However, our data demonstrate that PK1 knockdown does not disrupt the planar-polarized distribution of VANGL2, ROCK1, or F-actin at mediolateral junctions in the junctional zone, nor does it affect the formation of PCP-dependent supracellular actin cables[18] (Fig. 2). Instead, PK1 is specifically enriched at the apical cortex of medial junctional zone cells, where it is essential for the localized accumulation of F-actin and activated myosin - key drivers of apical constriction (Fig. 7). This functional uncoupling of PK1 from canonical Planar Cell Polarization reveals that, within the junctional zone, PK1 drives morphogenetic cell behaviors through a distinct mechanism. Our high resolution imaging confirms that apical F-actin accumulation is PK1-dependent in medial junctional zone cells (Fig. 7),

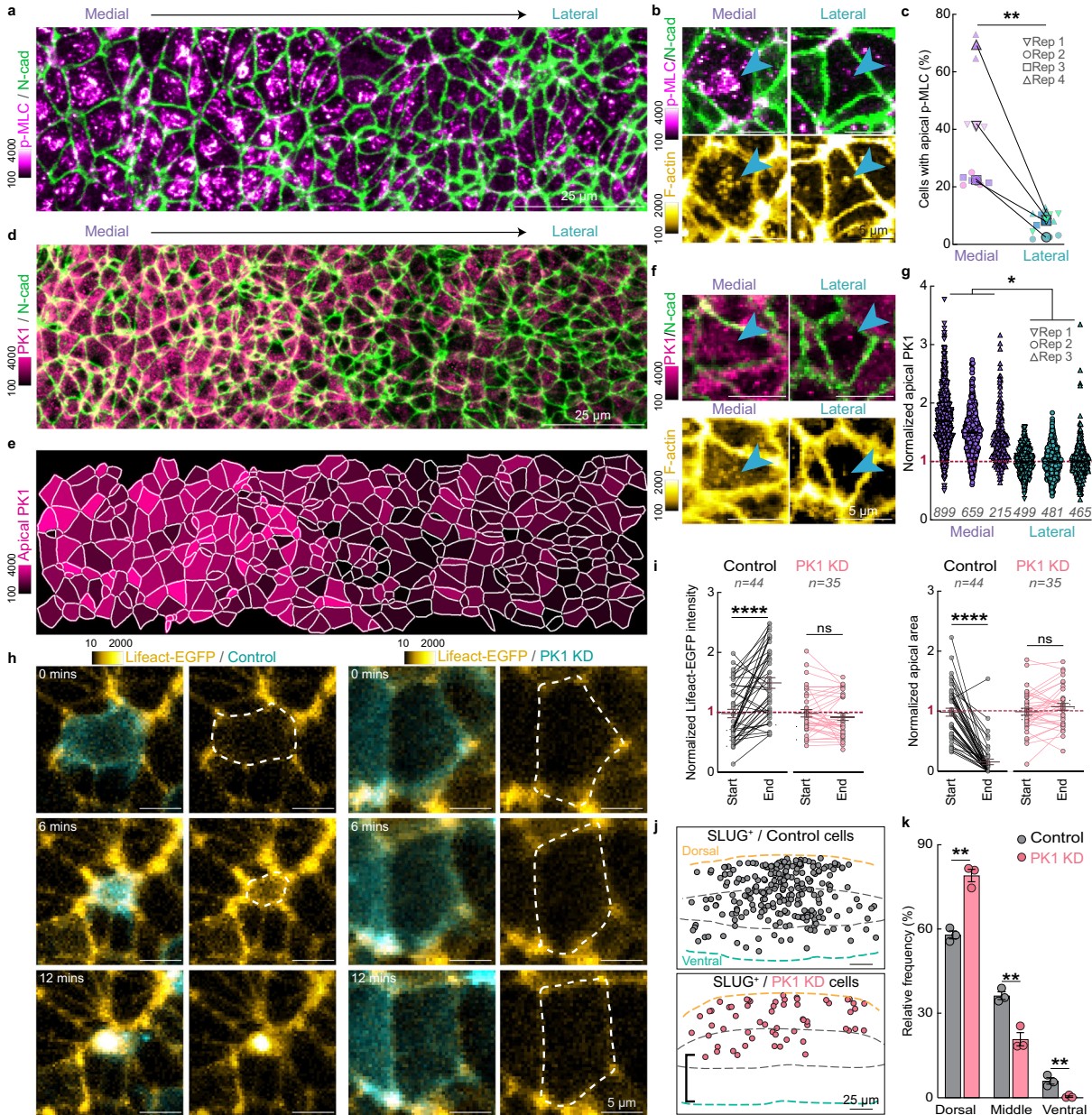

**Fig. 7 | PK1 regulates cortical actomyosin for apical constriction.**
**a** Immunofluorescent staining showing the differential distribution of apical p-MLC (magenta) across the mediolateral axis of the JZ. **b** Apically enriched p-MLC is observed in medial but not in lateral cells (cyan arrow heads, upper panel). The cell with apically enriched p-MLC in the medial region also shows enrichment of apical F-actin (cyan arrowheads, lower panel). **c** Comparison of the percentage of cells with apical p-MLC accumulation between medial and lateral region (n = 4 embryos, 2 ROIs per embryo), p = 0.0072, two-sided paired *t*-test. **d** Immunofluorescent staining showing PK1 localisation at the apical cortex of cells in the JZ. **e** Heatmap of PK1 intensity at the apical cortex in the JZ. **f**, **g** Medial cells have significantly more PK1 at the apical cortex than lateral cells, mean, ± sem (blue arrowheads, n = 3 embryos), p = 0.029, two-sided paired *t*-test. **h** Live imaging shows Lifeact-EGFP accumulates at the apical cortex of a Control cell undergoing ingression but not in a

PK1 KD cell. **i** Quantification showing Lifeact-EGFP significantly increases at the apical cortex of Control cells as they shrink their apical area, mean, ± sem (n = 44 cells from 3 embryos). PK1 KD cells (n = 35 cells from 3 embryos) show no change in Lifeact-EGFP intensity or apical area, mean, ± sem, p = 4.9 × 10⁻⁷ and 0.2461 for start vs end Lifeact-EGFP intensity in Control and PK1 KD cells, respectively; p = 9 × 10⁻¹⁵ and 0.0732 for start vs end apical area in Control and PK1 KD cells, respectively, two-sided paired *t*-test. **j**, **k** PK1 KD results in a significantly higher frequency of SLUG⁺ cells in the dorsal third of the JZ and significantly fewer SLUG⁺ cells in the middle and ventral thirds of the JZ compared to Control SLUG⁺ cells, (n = 314 cells from 3 Control embryos and n = 164 cells from 3 PK1 KD embryos), mean, ± sem, p = 0.0013, 0.0047, 0.0086 for dorsal, middle and ventral, respectively, two-sided un-paired *t*-test. Color scales depict pixel intensity [a.u].

suggesting that PK1 directly modulates cortical actin dynamics, as recently described in *Xenopus*[43]. Given that FGF signalling does not regulate PK1 expression in avian embryos[50], we propose that PK1 functions in parallel with the EMT program, coupling actomyosin–driven apical constriction to cell ingression.

PK1 is broadly expressed throughout vertebrate embryogenesis and studies in mouse, *Xenopus*, zebrafish and avian embryos have

demonstrated its requirement in neurulation and the development of cardiac, respiratory, renal, skeletal, dental, auditory, and ocular structures[51]. In frogs, fish, and ascidians, PK1 has been closely linked to convergent extension movements, reinforcing its classification as a core component of the PCP pathway. Accordingly, disruptions in these diverse developmental processes have typically been attributed to defects in planar cell polarity. However, analysis of the Pk1 mouse

mutant revealed a distinct phenotype characterized by defects in apicobasal polarity rather than in PCP, indicating that PK1 can function outside of canonical planar polarity[52]. Consistent with this, during *Xenopus* neurulation, PCP proteins including Prickle and Vangl are required for apical actomyosin accumulation and neural plate bending, implicating PCP components directly in the regulation of apical constriction rather than exclusively in mediolateral cell intercalation[53,54]. More recent work has provided compelling evidence for PCP-independent functions of Vangl2 in epithelial organization and tissue morphogenesis[55,56]. Together with growing evidence of PCP protein functions independent of planar polarity[43,57,58], these findings support an emerging view that PK1 promotes apical constriction as a key morphogenetic process. Our data extend this paradigm by demonstrating that PK1 functions independently of planar cell polarization in the junctional zone, where it drives apical constriction and medial cell ingression, suggesting that PK1 may regulate actomyosin-dependent epithelial remodeling in multiple developmental contexts beyond junctional neurulation.

The planar polarization-independent role of PK1 in the junctional zone may explain why PK1 disruptions lead to localized NTDs specifically at the site of medial cell ingression, in contrast to mutations in other core PCP proteins such as CELSR1-3 and VANGL1/2 which can impair neural tube closure along the anteroposterior axis[22–25]. We propose that the primary defect following PK1 disruption in the junctional zone is impaired ingression of medial cells. This conclusion is supported by our finding that targeted inhibition of actomyosin contractility confined to the medial region is sufficient to block lateral convergence and induce JNTDs (Fig. 5). Therefore, the defects observed in lateral convergence likely arise as a secondary consequence of the failure to clear medial cells from the junctional zone. Together, our findings not only advance our understanding of the cellular choreography underlying junctional neurulation but also provide mechanistic insight into the etiology of human junctional neural tube defects.

## Methods

### Experimental model and subject details

**Quail embryos.** Wildtype and transgenic quails were hosted and bred at the University of Queensland according to local policies approved by the University of Queensland Health Sciences Animal Ethics Committee (2023/AE000559). Fertilized quail eggs were stored in a 14 °C fridge and incubated at 37.5 °C with 60% humidity. Only fresh eggs laid within 1 week were used for experiments. Sex as a variable is not relevant at the stages studied.

### Method details

**Quail embryo EC culture.** Fertilized quail eggs were incubated for 20 to 30 h in a 37.5 °C humid environmental chamber to reach the desired stages based on the Hamilton and Hamburger (HH) stage chart modified specifically for quail embryos[59,60]. The quail eggs were cooled for approximately 1 h at room temperature (RT) before EC culture. The cooled quail egg was settled horizontally onto a customised egg holder. 0.5 mL of thin albumin was aspirated out from the blunt end of the egg with a 5 mL syringe (Nipro) to lower the embryo. A 1.0 × 0.5 cm window was opened on the top of the eggshell and the egg was then decanted into a 35 mm petri dish with the embryo resting on the top of the yolk. The albumin covering the surface of the embryo was carefully wiped away with Kimwipe tissue without damaging the embryo. A piece of 2.0 × 2.0 cm square Whatman filter paper with a 0.5 × 1.0 cm hole in the centre was placed onto the surface of the embryo. Care was taken to frame the embryo in the center of the hole. The filter paper was then cut along the edge with a pair of fine scissors (Fine Science Tools) to free the embryo from its surrounding tissues. The filter paper carrying the embryo was then elevated away from the yolk with fine forceps and submerged into Hanks' Balanced Salt solution (HBSS) buffer (Thermo Fisher Scientific, 14065056). Residual yolk was washed away from the embryo with a plastic transfer pipette and the filter paper carrying the embryo was then transferred into a clean 35-mm 1-well dish pre-coated with 1000 μL agar-quail albumin mixture composed of 0.3% weight/volume bacto-agar (Becton Dickinson, 214010) and 50% volume/volume quail albumin. Before further usage, the 1-well dish containing the embryo was kept in a humidified environmental chamber at RT.

**Quail embryo *in ovo* electroporation.** In the current study, a modified *in ovo* electroporation approach was utilized to transfect the cells in the JZ[61]. A 250 μm diameter platinum wire served as the cathode and a tungsten needle (Fine Science Tools, 10130-05) with 1 μm diameter at the tip and 125 μm diameter at the stem was used as the anode. Before the electroporation, the quail eggs were incubated to approximately HH8 when junctional neurulation begins and windowed as described above. Several drops of HBSS solution were applied onto the surface of the embryo to increase the electrical conductivity. A tiny hole was made at the boundary of the area opaca and area pellucida with a pair of fine scissors, through which the platinum cathode was inserted and placed underneath the JZ. A glass capillary (Drummond) was pre-pulled with a capillary puller (Sutter Instrument Co., model P-80 / PC) and the tip of the capillary was broken with a pair of forceps. The glass capillary was loaded onto a CellTram Oil microinjector (Eppendorf South Pacific, 5196000030) and the plasmid mixed with sucrose solution and Fast Green FCF indicator solution (Sigma Aldrich, F7252-5G) (final concentration for DNA, sucrose and indicator solution was 0.5-3.0 μg/μL, 5% and 0.05%, respectively) was loaded into the capillary. The capillary tip penetrated the vitelline membrane and was positioned above the JZ. Approximately 0.1 μL of the DNA mixture was injected into the interstitial space between the JZ and vitelline membrane with the oil microinjector, and the tungsten anode was instantly positioned close to the JZ without contacting the tissue. A series of square electrical pulses (three pulses, 5 volts per pulse with 25-msec pulse length and 900-msec resting time,) was applied and the electrode and the glass capillary were removed from the manipulated embryo. The electroporated embryo was left at RT for recovery for approximately 3 h and placed back in the 37.5 °C humidified environmental chamber to develop to the desired stage.

**Quail embryo *ex ovo* electroporation.** Embryos were first incubated to the desired stage and cultured by EC culture. The plasmids or morpholinos were prepared and loaded into the micro-injection setup as described above. The negative electrode (Nepagene, CUY701P2E) is a 60 mm petri dish with a platinum foil sheet (4 mm diameter) embedded in the centre. The positive electrode (Nepagene, CUY701P2L) is a hand-held stem with a flat platinum paddle (4 mm diameter) at the tip. Both negative and positive electrodes were connected to the electroporator (Super Electroporator NEPA21) by platinum wires. Before electroporation, the negative electrode (in the petri dish) was filled with around 30-50 mL HBSS solution and the depth of the solution was around 5–7 mm. The embryos were then submerged into the HBSS with the ventral side up. The tip of the capillary penetrated the embryonic ventral tissue and approximately 0.1 μL of the plasmid or morpholino mixture was injected underneath the node area but above the vitelline membrane with the oil microinjector. A series of square electrical pulses (five pulses, 5.2 volts per pulse with 50-msec pulse length and 200-msec resting time) were applied and the embryos were picked up from the petri dish and positioned in the 35 mm 1-well dish pre-coated with agar-quail albumin mixture. The electroporated embryos were allowed to recover at RT for around 3 h before they were returned to the 37.5 °C humidified environmental chamber to develop to the desired stage.

**Whole-mount quail embryo immunofluorescent staining.** Embryos were fixed with 4% paraformaldehyde (PFA, ProSciTech, C00432-1010) at RT for 30 min and subsequently washed 3 times with phosphate-buffered saline (PBS) at RT, 5 min each time. To increase the penetration of the antibodies, the fixed embryos were permeabilized in PBS containing 0.5% Triton X-100 (Sigma-Aldrich, T9284-500mL) at RT for 3 times, 30 min each time. The embryos were then blocked with blocking buffer for 3 h at RT. The blocking buffer was composed of 0.5% Triton X-100, 0.02% sodium dodecyl sulphate (SDS, Sigma-Aldrich, L4509-25G) and 1% bovine serum albumin (BSA, Sigma-Aldrich, A2153-50G) in PBS. After blocking, the embryos were submerged in primary antibodies diluted in dilution buffer (PBS containing 0.5% Triton X-100, 0.02 % SDS and 0.2 % BSA) overnight at 4 °C, and the unbound antibodies were washed away with PBS containing 0.5 % Triton X-100 (0.5 % PBST) 6 times at RT, 10 min each time. Primary antibodies used were: SOX2 (Abcam, AB97959, 1:200), PRICKLE1 (Proteintech, 22589-1-AP, 1:200), ZO-1 (Thermo Fisher Scientific, 33-9100 and 40-2200, 1;400), VANGL2 (Sigma Aldrich, MABN750, 1:50), ROCK1 (ABClonal, A11158, 1:100), p-MLC (ABClonal, AP1433, 1:100), Fibronectin (DSHB, B3/D6-s, 1:4), Fibronectin (Sigma Aldrich, F3648, 1:100), SLUG (New England Biolabs, 9585 T, 1:100), N-Cadherin (DSHB, 6B3-s, 1:100), E-Cadherin (BD Biosciences, 610181, 1:100). After primary antibody incubation, the embryos were then incubated with the corresponding secondary antibodies (Alexa Fluor™, Invitrogen) diluted in the dilution buffer (1:1000) overnight at 4 °C and the unbound antibodies were washed away with PBST 6 times at RT, 10 min each time. Depending on the purpose of the staining experiment, embryos could also be counterstained with DAPI (Sigma-Aldrich, MBD0015-15ML, 1:2000 diluted in 0.5 % PBST) for 15 min at RT to visualise the nuclei and/or with phalloidin (Abcam, 1:1000 diluted in 0.5 % PBST) overnight at 4 °C to visualise the F-actin. The embryos were then mounted with mounting media (Sigma-Aldrich, F4680-25ML) on a six-well glass bottom plate (Cellvis, P06-1.5H-N) for imaging or stored at 4 °C in PBST for further usage.

**PRICKLE1 knockdown.** An enhanced version of miRNA knockdown vector named spliced microRNA155 (pSM155) was a generous gift[62]. A customised miRNA expression cassette was cloned into an artificial intron upstream of a fluorescent protein reporter. Compared to a conventional miRNA system, this unique design increases the expression of both knockdown miRNAs and fluorescent protein because the miRNA is spliced out, leaving an intact 5′ CAP-exon-EGFP-3′ poly(A) component[30,62]. miRNA targeting quail *PRICKLE1* mRNA (XM_015854695.2, the targeted region is 5′- AGT TTC GGG CCT ACA TTC AAA-3′) was purchased from Integrated DNA Technologies. The top oligo sequence of the miRNA was 5′- TGCTG TTT GAA TGT AGG CCC GAA ACT GTT TTG GCC ACT GAC TGA CAG TTT CGG CTA CAT TCA AA-3′; the bottom oligo sequence was 5′ CCTG TTT GAA TGT AGC CGA AAC TGT CAG TCA GTG GCC AAA ACA GTT TCG GGC CTA CAT TCA AAC −3′ (the 5′ overhang is underlined). The top oligo sequence of a scrambled control miRNA was 5′-TGCTG TTT GAA TGT AGG CCC GAA ACT GTT TTG GCC ACT GAC TGA CAG TTT CGG CTA CAT TCA AA-3′; The bottom oligo sequence was 5′-CCTG TTT GAA TGT AGC CGA AAC TGT CAG TCA GTG GCC AAA ACA GTT TCG GGC CTA CAT TCA AAC-3′ (the 5′ overhang is underlined). The knockdown or scrambled control miRNAs were inserted between two *BsmBI* sites of pSM155 system according to the previously published methods[62]. The sequences of siRNAs were as follows: Quail PK1 siRNA: 5′-CAUGG-GAACUCUGAAUUCU (dTdT)−3′; Control scrambled siRNA: 5′-GAAUUGCGGCAUCAUACUU (dTdT)−3′. 1 µg/µl siRNA was co-electroporated with 0.5-1 µg/µl H2B-miRFP670.

**SLUG knockdown morpholino design and application.** A translation-blocking morpholino sequence (5′- GAA CGC AGG CTT TCT CCT TCT TGC A-3′) complementary to the 5′ UTR of quail SLUG and a standard negative control sequence (5′-CCT CTT ACC TCA GTT ACA ATT TAT A-3′) were ordered from GENE TOOLS, LLC. A carboxyfluorescein was tagged to the 3′ to charge the morpholino for electroporation. The lyophilised morpholino was resuspended to 5 mM with nuclease-free water and stored at −20 °C. Before electroporation, the morpholino solution was mixed with sucrose, Fast Green and DNA plasmid as a conductive carrier. The final mixture was composed of 0.2-2 mM morpholino, 6% sucrose, 0.05% Fast Green and 0.5-2 mg/mL DNA. The morpholino mixture was applied to the JZ for either *in ovo* or *ex ovo* electroporation. Since the morpholino is tagged with fluorescein, morpholino uptake could be checked immediately after electroporation using a fluorescent stereomicroscope. The successfully electroporated embryos were selected for further experiments.

**FGF receptor inhibitor application.** The pan-FGF receptor inhibitor infigratinib (also called NVP-BGJ398, Focus Bioscience, 10 mM in DMSO) and DMSO control were diluted 1:1000 in HBSS. For cell ingression analysis, 20 µL infigratinib or DMSO control was applied above and underneath the EC cultured embryo. The embryo was then left at RT for 3 h and subsequently electroporated with H2B-tagged fluorescent protein to label the nuclei by *ex ovo* electroporation. After electroporation, infigratinib or DSMO control was re-applied, and the embryo was returned to the 37.5 °C humidified environmental chamber to develop to the desired stage. The embryos were then fixed and used for cell ingression analysis.

**Azido-blebbistatin activation.** TgT2(CAG:NLS-mCherry-IRES-GFP-CAAX) quail embryos were cultured out and placed into a glass-bottomed plate pre-coated with 250 µL agar-quail albumin mixture. Azido-blebbistatin (100 mM in DMSO, Motorpharma) was diluted 1:10,000 in HBSS with a final concentration of 10 µM and 1:10,000 diluted DMSO was used as the negative control. 20 µL of Azido-blebbistatin or DMSO was applied above and underneath the embryos, and the embryos were incubated for 1 h at 37.5 °C. Photoactivation was performed with a Zeiss LSM710 confocal microscope equipped with Plan Apochromat, 20 × / 0.8 numerical aperture objective. A region of interest (ROI) surrounding the medial JZ was drawn with the free hand tool in Zen software and exposed to 1.5−2 min irradiation with 30% 405 nm laser. The embryo was returned to 37.5 °C and the activation process was repeated 1.5 h later. After photoactivation, the irradiated embryos were incubated overnight at 37.5 °C and fixed in 4% PFA. The width of the posterior neuropore was then measured with Fiji and the difference in the width between Azido-blebbistatin and DMSO group was quantified in PRISM.

**Confocal imaging of fixed embryos.** Mounted embryos were imaged with a Zeiss LSM 710 Confocal Scanner with LD C-Apochromat 40 × / 1.1 numerical aperture long working distancing objective. The image was acquired at 1024 × 1024 resolution with a scanning speed of 3.15 µsec/pixel. Appropriate z-directional thickness was set to cover the ROI. The 488 nm and 561 nm channels were imaged with 488 nm and 561 nm GaAsP detectors. The remaining channels were imaged with built-in detectors in LSM 710.

**Live imaging.** EC cultured quail embryos were put into a glass-bottomed plate pre-coated with 250 µL agar-quail albumin mixture. The embryos were usually placed with the dorsal side down. For live epifluorescence microscopy, the embryos were imaged with a Zeiss Axio Observer 7 inverted Microscope equipped with a Plan Apochromat 10×/0.45 numerical aperture objective. Sufficient tiles and z-directional thickness were set to cover the ROI. The scanning interval was approximately 30 min. For live confocal imaging, embryos were imaged with a LSM 710 Confocal Scanner with LD C-Apochromat 40 × /1.1 numerical aperture long working distancing objective. The 488 nm and 561 nm channels were imaged with 488 nm and 561 nm GaAsP

detectors. The remaining channels were imaged with built-in detectors in LSM 710. The image was acquired at 512 × 512 resolution with a scanning speed of 3.15 µsec/pixel and sufficient tiles and z-stack to capture the ROI. Depending on the purpose of the experiment, the scanning interval was usually set between 5–10 min. Embryos with normal development were used for further quantitative analysis.

### Quantification and statistical analysis

**Quantification of junctional NT transformation.** Cultured Lifeact-EGFP embryos were live-imaged with a Zeiss epi-fluorescent microscope and the live imaging data was imported into Imaris software (Bitplane, version 10.2.0, imaris.oxinst.com). All movies were registered based on the position of the 5[th] somite with *Reference* function in Imaris. The anterior border of the junctional NT was defined as the Hensen's node, and the posterior border was defined as the anterior tip of the primitive streak. The width was defined as the ML distance between the bilateral neural plate across the node and the length was defined as the AP distance between the anterior and posterior border of the junctional NT. The width and the length during junctional neurulation were exported from Imaris and were plotted against time in GraphPad PRISM.

**H2B-based cell tracking and cell migration parameters quantification.** TgT2(CAG:NLS-mCherry-IRES-GFP-CAAX) quail embryos were electroporated *in ovo* with H2B-miRFP670. Embryos were then cultured, and live imaging was performed by confocal microscopy. Two channels were acquired: 561 nm for nuclear mCherry and 647 nm for H2B-miRFP670. Images were stitched and the drift was corrected in Imaris with the *Reference* function. The *Spots* function in Imaris recognised the H2B-miRFP670 signal in each time frame and linked the sequential frames to generate continuous cell trajectories.

The X and Z directional displacement of cells was automatically calculated by Imaris. The JZ on either side of the midline was divided into three equal sections. Cells located in the innermost third, closest to the midline, were classified as medial. Cells in the outermost thirds, farthest from the midline, were classified as lateral. The X and Z displacement was then compared between the medial and lateral cells in PRISM.

For cell directionality analysis, the 3D location of each track across time was exported from Imaris and imported into MatLab. The starting location was subtracted from each timepoint of the trajectory. After this processing, each trajectory started from [0,0,0] in 3D coordinates. The 3D trajectory was then projected into 2 types of 2D trajectory: the in-plane trajectory using X and Y locations and the transverse trajectory using X and Z locations. The 2D trajectory was binned every 20° to generate the polar histogram. The polar histogram was then weighted using each bin's average trajectory displacement.

**Quantification of cell ingression in fixed embryos.** Wildtype quail embryos *ex ovo* electroporated with H2B-tagged fluorescent protein were incubated to develop to the desired stage. The embryos were then fixed, stained with DAPI and phalloidin and imaged by confocal microscopy. Across the transverse plane of the JZ, the distance of the H2B-labelled cell to the dorsal ("D") and ventral layers ("V") was measured manually in Fiji. The relative distance of the H2B-labelled cell to the dorsal surface is defined as D/(D + V) × 100%. For each embryo, approximately 10 transverse planes were sampled along the anterior-to-posterior axis within the JZ. In GraphPad PRISM, the relative distance was binned every 10% to generate the histogram.

**Quantification of SLUG expression level.** Fixed embryos were stained with SLUG antibody and DAPI and imaged with a confocal microscope. Across the transverse plane, the DAPI signal was segmented with Cellpose, a machine learning-based cell segmentation software[63]. The segmented result was imported under the *Surface*

function in Imaris. The SLUG signal was then masked using the nuclear segmentation and the SLUG background was moved to a separate channel. The mean intensity of the background SLUG channel was automatically determined by Imaris. The mean intensity of background SLUG + 2 standard deviation was then used as the threshold to determine whether a cell is SLUG positive. A nucleus is classified as SLUG-positive if its average SLUG intensity exceeds the mean background SLUG intensity by more than two standard deviations. Conversely, it is classified as SLUG-negative if its average SLUG intensity is less than the mean background SLUG intensity plus two standard deviations. This approach enabled the measurement of SLUG-positive cells to be determined in an automatic and unbiased manner. For each embryo, approximately 10 transverse planes were sampled along the anterior-to-posterior axis within the JZ. When quantifying SLUG morpholino knockdown efficiency, only the cells successfully transfected with control or SLUG morpholino were analysed.

**Quantification of PRICKLE1 expression level.** Fixed embryos were stained with PK1 and ZO-1 or N-cad antibodies and imaged with a confocal microscope. For baseline PK1 quantification at 5 and 9-somite stage, the image was projected into 2D using the ZO-1 or N-cad channel with LocalZProjector, a Fiji plugin assuring accurate 2D projection in epithelia with complex topology[44]. The expression level of PK1 in each cell was then quantified using a customised workflow. Firstly, the cell outline visualised by the ZO-1 or N-cad signal was auto-segmented, followed by manual correction with Tissue Analyzer, another FIJI plugin[64]. Secondly, the segmented binary result was imported into MatLab where the mean PK1 intensity in each cell was extracted. PK1 intensity between the primary NT and JZ was further compared in GraphPad PRISM. For analysis of PK1 knockdown efficiency, only the cells successfully electroporated with PK1 knockdown or scrambled control plasmid were selected. The PK1 intensity was then normalized by the ZO-1 intensity (ZO-1 intensity has been quantified to show no difference across PK1 knockdown and control embryos in the current study). PK1 intensity between the knockdown and control embryo was further compared in GraphPad PRISM.

**Quantification of PRICKLE1 and PCP polarity.** Embryos stained with antibodies against PCP proteins were imaged with a confocal microscope. TgT2(CAG:NLS-mCherry-IRES-GFP-CAAX) embryos were fixed and the membrane EGFP signal was used as the negative control. The image was projected into 2D with LocalZProjector and ML and AP-aligned junctions sharing a vertex were overdrawn using the segmented line tool in Fiji. Junctions with an angle of less than 45° relative to the medial-to-lateral axis of the JZ were identified as ML junctions, and junctions with an angle of more than 45° were identified as AP junctions[65]. Polarity was quantified by calculating the ratio of PCP protein staining intensity along the ML axis to that along the AP axis. A ratio greater than one indicates that the protein is enriched at ML junctions, a ratio less than one indicates AP junction enrichment, and a ratio equal to one suggests no polarity[65]. Overall, 4-6 ROIs were chosen per embryo. In GraphPad PRISM, the polarity was compared between the PCP proteins and membrane EGFP in wildtype, PK1 knockdown and scrambled control cells.

**Quantification of cell apical area in both fixed and live embryos.** For static analysis of cell apical area after PK1 knockdown, Lifeact-EGFP embryos were electroporated with either scrambled control or PK1 knockdown miRNA expression vector. The embryos were then returned to 37.5 °C to develop to the 7-8 somite stage at the midpoint of junctional neurulation. The embryos were then EC cultured and imaged with a confocal microscope at one static timepoint. Based on the Lifeact-EGFP signal, cell apical surface was projected into 2D with LocalZProjector and the cell boundaries were then segmented with Tissue Analyzer. The area of each cell was automatically quantified by

Tissue Analyzer. The average apical area of scrambled control or PK1 knockdown cells were compared in GraphPad PRISM.

For dynamic analysis of cell apical area after PK1 knockdown, Lifeact-EGFP embryos electroporated with either scrambled control or PK1 knockdown miRNA expression vector were live imaged at an interval of 5–10 min for up to 3 h. The cell apical surface was then projected into 2D with LocalZProjector and the cell outline in each frame was drawn using the free-hand tool in Fiji. The cell area was then quantified in Fiji and the result was exported to Excel for further processing. In Excel, the cell area over time was normalized by its largest area and all the cells were synchronised by the timepoint when they reached their maximal area[42]. The apical area of all cells in each timepoint was averaged, and the average apical area between control and PK1 knockdown cells was then compared over time in GraphPad PRISM.

**Quantification of PRICKLE1 and actomyosin apical enrichment.** Wildtype embryos were stained with antibodies against PK1 and p-MLC. Embryos were also stained with phalloidin to visualise the F-actin network. To facilitate further segmentation, embryos were also co-stained with ZO-1. The cell apical surface (approximately 10 μm of the dorsal side of the tissue) was imaged with a LSM 710 confocal microscope with built-in Airy-scanning mode using Plan Apochromat 40 × /1.3 numerical aperture oil objective. To selectively visualise the apical surface, only the apical domain (around 3 μm thickness) was extracted with LocalZProjector using the ZO-1 signal[42].

To quantify the apical PK1 intensity, cells were segmented with Tissue Analyzer. The segmented result was further imported into MatLab. In Matlab, a custom script identified the segmented cell boundary and transformed the cell boundary location information from Cartesian coordinates into polar coordinates. The cell cortex was then defined as an area composed of 80% of the entire radius from the cell centroid[26]. This definition successfully separates the cell apical cortex from the cell boundary. The mean PK1 intensity within the cell cortex was calculated and the difference in cortical PK1 intensity between medial and lateral cells was compared in GraphPad PRISM.

To quantify the p-MLC apical enrichment, the percentage of cells with apically accumulated p-MLC was manually calculated between the medial and lateral areas. 3-5 ROIs in medial and lateral areas per embryo were chosen and the average percentage in medial and lateral regions was compared in GraphPad PRISM.

To quantify the cortical F-actin change across time after PK1 knockdown, knockdown and control cells were first tracked manually in Fiji. The cortical area was manually drawn in the first and last timepoint for each track with the free-hand tool in Fiji. The intensity of cortical F-actin was then measured by Fiji. In GraphPad PRISM, the cortical F-actin change between the first and last timepoint was compared in both control and PK1 knockdown cells.

**Statistics and reproducibility.** All experiments were independently repeated at least three times with similar results. One quail embryo was defined as a biological replicate ($n = 1$). Embryos were randomly allocated to experimental groups. Representative images shown in the figures are from experiments that were replicated independently a minimum of three times with comparable outcomes. Quantitative data is presented as mean ± sem and all the statistical analysis was conducted with Graphpad PRISM software (version 10.2.0). For all the experiment, ns, $P > 0.05$; *$P < 0.05$; **$P < 0.01$; ***$P < 0.001$; ****$P < 0.0001$. Unpaired $t$-test was used when comparing two populations. Paired $t$-test was used when comparing two sampling results taken from the same embryo. In a few cases, the Kolmogorov–Smirnov test was performed to compare the differences in data distribution between two populations. The details of the statistical analysis for specific experiments are described in the corresponding Figure legends.

**Reporting summary**

Further information on research design is available in the Nature Portfolio Reporting Summary linked to this article.

## Data availability

The data generated in this study are provided in the Source Data file. The raw imaging data underlying the figures have been deposited at The University of Queensland's institutional repository, UQ eSpace, and are publicly available[66]. Source data are provided with this paper.

## Code availability

The codes generated in this study have been deposited at The University of Queensland's institutional repository, UQ eSpace, and are publicly available[66].

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

## Acknowledgements

Microscopy was performed at the Institute for Molecular Bioscience Advanced Imaging Platform. We thank the IMB Microscopy facility staff. M.D.W was supported by a Future Fellowship (FT200100899) and a Discovery Project grant (DP220101878) from the Australian Research Council (ARC), and Ideas Grants (2013027, 2038843) from the National Health and Medical Research Council of Australia (NHMRC). S.J.S was supported by a Future Fellowship (FT190100516) from the Australian Research Council (ARC).

## Author contributions

J.X.W.: investigation, formal analysis, methodology, software, data curation, visualization, writing—review and editing. Y.D.A.: formal analysis, methodology, software, data curation, writing—review and editing. S.Z.T.: methodology, writing—review and editing. S.N.R.: investigation. S.J.S.: writing—review and editing. M.D.W.: conceptualization, data curation, formal analysis, funding acquisition, methodology, project administration, supervision, validation, visualization, writing—original draft preparation, writing—review and editing.

## Competing interests

The authors declare no competing interests.
