## [Transparent Peer Review file · Nature Communications]

Quantitative live imaging reveals PRICKLE1 controls junctional neural tube morphogenesis independent of Planar Cell Polarity

Corresponding Author: Dr Melanie White

Version 0:

Reviewer comments:

Reviewer #1

(Remarks to the Author)

I rather like this interesting new paper that not only characterizes an element of neural morphogenesis that has gone all but ignored but finds some interesting new roles for the PCP protein Pk1.

The paper focuses on “junctional neurulation,” the process that is important for linking primary and secondary neurulation in amniote embryos (and for potential clinical relevance) but has received scant attention. Using transgenic quail, the authors very nicely describe the cell behaviors underlying the process, notably showing that apical constriction, EMT, and ingression in the medial region of the neural epithelium are key players. Pk1 had previously been shown to be important for junctional neurulation, and that is confirmed and extended substantially here. Live imaging reveals that Pk1 is essential for apical constriction in these cells. Surprisingly, however, while these cells are planar polarized and seem to be engaged in rosette-mediated convergent extension, pk1 loss does not affect either (but see minor comment 2, below).

I strongly support publication in Nature Communications, both for the detailed characterization of an important but ignored aspect of biology and for the analysis of Pk1. (Indeed, “PCP-independent” roles for PCP proteins are an emerging point of interest.) There is a huge amount of data here and I think it imperative that their publication not be delayed by reviewers who -no doubt- will provide a laundry list of other experiments that might be done. These might even ‘improve’ the paper, but I think it already represents a substantial advance and calls for some intangible “more” should be refused.

I will however make one major comment that must be addressed in some way, as it relates directly to the veracity of the data. All other comments are mere suggestions.

Major comment:

In the paper, both the Pk1 miRNA and the Slug MO lack specificity controls to rule out off-target effects. The standard in the field for knockdowns in fish and frogs would be to include controls like rescue experiments or phenocopy with an orthogonal reagent (e.g. the same reagent targeting a distinct sequence, or a distinct reagent like a dominant-negative fragment – of which there are many of Pk1).

I do not know the standards in the avian biology community, and it does seem to me that the fly and worm communities use siRNA quite liberally without such controls. But I feel that either off-target controls must be added to the paper, or it must be very explicitly stated in the results section that off-target effects cannot be ruled out.

Minor comments:

1. The authors might be more careful/explicit about what they mean by “PCP-independent.” PCP, after all, describes both the signaling pathway and the cellular property. By showing polarization of Vangl remains intact after Pk1 KD, it seems they are right with regards to cellular property. But because they did not disrupt any other PCP protein, Pk1 yet may require the entire signaling pathway to effect ingression.
2. Is there really convergent extension in junctional neurulation? Rosettes are shown in Supp Fig. 2D but are not quantified for controls or KDs. The simpler T1 transition are also not quantified. This oversight is important because the difference in AP/ and ML junction dynamics in Supp. Fig. 2E, F seem barely significant. Is this tissue really doing convergent extension?

A better way to ask the question is to quantify the number of effective event for rosette formation and separately for rosette resolution. As well as the same for T1s. Then ask, does Pk1 change these?

I think these things can be quite easily extracted from the data already in hand and could be important. Such data would help us ask 1) if C?E happens here, 2) if Pk1 controls it and 3) and maybe PCP does control it but without Pk1 (i.e. via Pk2).

3. In Fig. 1I, it is not clear to me that these images are sufficient to back the claim that there is more Pk1 in these cells. I do not feel that normalizing to actin is sufficient to rule out issues like cell density here. This is important, too, because this assay is the only control to show the Pk1 knockdown is effective. Ideally, I'd like to see a Pk1 western blot to show efficacy of the KD.

4. How is it possible the distributions in fig. 2C are so hugely significant, but the aggregate difference in fig. 2dis barely significant? I am confused.

5. The literature in this field is vast and complex. I hope the reviewers will consider the following suggestions concerning their treatment of it.

5a. Citation 17 (Blankenship) is inappropriate. That paper says nothing of PCP proteins and indeed deals with *Drosophila* CE, which is independent of PCP proteins.

5b. Citation 19 is fine, but this connection between PCP, CE, and neural tube closure was shown first by Wallingford and Harland ten years prior (Development 2002).

5c. It seems a shame the Discussion does not consider the several papers from the Sokol Lab on PCP in apical constriction during neurulation (e.g. papers by Ossipova in 2014 and 2015; there are others).

5d. Likewise, the curious results here are lent credence by recent work from the Devenport lab on PCP-independent roles for Vangl2 (papers last year from Paramore in Dev Cell and Development).

5e. It is relevant to this paper's findings that while Pk1 was linked to C/E in frogs, fish, and ascidians, the phenotype of the Pk1 mouse mutant was a defect not in PCP but in apical/basal polarity (Tao et al., 2009).

5f. Ref 6 seems to be missing the journal name and page numbers, etc.

Reviewer #2

(Remarks to the Author)

Summary: The manuscript, "Quantitative live imaging reveals PRICKLE1 controls junctional neural tube morphogenesis independent of Planar Cell Polarity," focuses on understanding the mechanisms driving the closure of the neural tube between the primary and secondary neurulation regions in the spinal cord of transgenic quail embryos. They identify that the PCP protein, Prickle1 (PK1), drives apical constriction via actomyosin accumulation and that it is necessary for junctional neural tube closure at the border where mediolateral convergence in primary neurulation and cell ingression during secondary neurulation occur. Overall, this is a very interesting paper, and the text conclusions are well supported by the data. My comments are below.

Comments:

1. In the introduction do the authors mean amniotes when they refer to higher vertebrates? If so, it might make it clearer to state that because fish and amphibian neurulation processes may differ from those described here.

2. Can the authors provide a black and white image of PK1 alone in addition to the overlay with DAPI in Figure 1? Adding a PK1 with Phalloidin overlay would be helpful here too.

3. I can see that the knockdown cells still move significantly less, but the cell convergence tracks in the control embryo in figure 2B are very different (less convergence) than the tracks in figure 1E, which seem very directed to the midline. Are these embryos different stages or is there an explanation for this difference?

4. The significance indicators (asterisks) are missing from the graph in figure 2C that shows displacement.

5. I like that in figure 4, the rationale and experimental design is great and supports the foundation that these cells are ingressing using traditional EMT mechanisms. Although this may be out of scope for this paper, it would be interesting to see how the localization of cell adhesion molecules changes in these cells as the adherens junctions are remodeled during PK1-mediated ingression. At these early stages E-cadherin is expressed in the neural ectoderm while N-cadherin is expressed in the underlying mesenchymal cells (Dady et al., 2012 and Rogers et al., 2018) and slug (in theory) may be linked to driving these changes to promote EMT. I would like to see cadherin localization changes in the PK1 knockdowns and the Slug knockdowns (figure 5 and figure 7). Also- are the adherens junctions intact in these embryos where the cells do not ingress (maintenance of cell-cell adhesion and lack of constriction)? Is the basement membrane still remodeled/removed in the midline although the cells cannot navigate in?

6. In figure 4G, the authors use fibronectin as a marker of the basement membrane, but that is not a usual marker of the basement membrane itself like laminin for example. They may want to alter the text here as fibronectin can be organized by the existing basement membrane but this would be an indirect readout.

7. In my opinion, the FGF blocking experiments don't really add much to the story and in fact- they may complicate it a bit. The Stern, Harland, and Bronner labs (among others) have shown that FGF signaling is required for neural plate and neural plate border induction and it is not clear that in the embryos treated with the FGF inhibitor (figure 5 D-H) that these tissues are specified appropriately. I would either remove these experiments, or I would show that SOX2 and neural plate border (PAX7) expression is normal in the embryos treated with the FGF inhibitors.

8. In figure 5I the embryo appears torn straight through in the midline rather than a basic failure of neural tube closure that we see in the other figures. It may just be the image provided but this phenotype does not appear to phenocopy the image in figure 1J. Can the authors comment on this or provide a more representative embryo?

9. I don't think that fibronectin and N-cadherin need to be capitalized in the text, but rather the authors can abbreviate them in the text (FN and N-cad).

10. What do the neural ectodermal cells that ingress become? I know figure 1D shows that some of them reincorporate into the neural tube- but how? Do others end up in the mesodermally-derived tissues? I would like to see a section of the embryo from 1D to demonstrate this. For those cells that don't reincorporate, do they die after ingression? Apoptosis at junction points is necessary for neural tube closure (Roellig et al., 2022) so it would be interesting to look at activated caspase or TUNEL in these cells in the wild type and PK1 perturbed embryos to see if these cells are undergoing true ingression or apoptotic mediated anoikis.

11. If there is space it would be nice to move Supp. Figure 3C to the main body of the paper. The morphology of the slug positive neuroepithelial lateral cell vs. the mesenchymal ingressing medial cell are very cool and demonstrate the differences in these regions really well.

Reviewer #3

(Remarks to the Author)

In this paper, Wang et al investigate the mechanisms underlying neural tube closure in an understudied region of the neural tube – the junctional zone – where primary neurulation and secondary neurulation meet. Using live and quantitative imaging of transgenic quail embryos they show that the junctional neural tube forms through mediolateral convergence and EMT-driven ingression, dependent on the PCP component PRICKLE1. They further present data suggesting that PK1 acts in a PCP-independent manner at the apical cortex of cells, where it drives actomyosin accumulation and apical constriction.

Overall, these findings are significant because they identify both a new cellular behavior critical for neural tube closure and the underlying molecular mechanism, in an emerging model system that is relevant for understanding human disease. They also define new and unexpected functions for PK1, a gene that is implicated in neural tube defects in humans. The authors support their conclusions with robustly quantified data generated through advanced live imaging techniques. The data is presented clearly, and most of the conclusions are consistent with their data. Notably, this work is done in a technically state-of-the-art system of transgenic quail embryos, with similarities to human neural tube formation not present in mouse. For these reasons this paper would be of broad interest to many, including cell biologists, developmental biologists focused on morphogenesis, and clinicians interested in structural birth defects.

Some conclusions, however, could be toned down or strengthened by additional evidence. A main conclusion of the paper is that PK1 acts independently of PCP in junctional neurulation, but the evidence is rather weak and would require additional controls and experimental data. Toning down this conclusion would also alleviate this concern, as I feel it is a minor point and wouldn't take away from the overall novelty and advance of the paper.

Major comments:

1. One point that needs further investigation regards the uncoupling of PK1 activity from classical PCP axis establishment. The authors conclude that PK1 functions independently of planar polarity cues by assessing VANGL2 localization as well as the length of mediolateral junctions. However without knowing how loss of Vangl2 affects neurulation and junction length, this data is hard to interpret. Another possibility is that PK1 is required downstream of Vangl2. Does VANGL2 knockdown result in a difference in length of mediolaterally oriented junctions? This should be added to justify the use of mediolateral junction length as a metric for planar polarity function in the junctional zone. Is PK1 localization also VANGL2 independent?

2. Can the authors elaborate about the relationship between EMT/ingression and mediolateral convergence? Does EMT-driven ingression of medial neuroepithelial cells contribute to the mediolateral convergence of lateral cells? Both processes are affected by PK1 KD, but are these two separate behaviors driven by two PK1-dependent processes or is convergence a passive consequence of ingression? Can the SLUG KD data be used to explore how inhibiting EMT affects mediolateral convergence? Alternatively, could the authors use azidoblebbistatin to inhibit myosin selectively in the medial zone and determine the impact on lateral cell movements?

3. Data showing PK1 subcellular localization could be improved as it is not clear from the images provided in Figure 7 that it is apical and cortical. The authors should include z-projection images to demonstrate an apical PK1 cap and characterize the subcellular localization of PK1 along the apical-basal axis. Also, what is the evidence that the PK1 antibody used is specific? Additionally, Figure 7B should include a staining for PK1 to demonstrate colocalization to the apical enrichment of F-actin and p-MLC.

4. Does loss of PK1 affect cell fate in the medial junctional zone? Quantification of the number of SLUG+ cells in the medial junctional zone of PK1 KD embryos is needed.

Other comments:

1. Figure 1I: Include PK1 KD embryos to demonstrate that the PK1 antibody is specific
2. Figure 1I: Include additional panels to characterize the subcellular localization of PK1
3. Figure 2F-G: Planar polarization should be quantified as ML junctional enrichment. Quantifying junctional intensity is not sufficient to demonstrate polarization. ML junctional intensity must be quantified relative to AP junctional intensity to demonstrate planar polarization.
4. Figure 3B: Can the authors include still frames from the H2BiRFP channel, zoomed in, to demonstrate relative positions of nuclei through time as they ingress? The cell tracks shown don't distinguish between the whole tissue moving ventrally versus individual cells internalizing. This is an important point of the paper and should be shown as convincingly as possible.
5. Figure 3C-J: Suggest removing the lined connected the data points. Lines suggest the x-axis is time, and the data would be clearer without the connections.
6. Figure 5A-C: The authors should validate that the SLUG MO cells in the medial junctional zone that remain dorsal have lost their mesenchymal or protrusive morphology.
7. Figure 5D-E: Given that the % of SLUG+ cells is still around 50% in Panel D, include a staining for SLUG expression in the FGF_i condition in Panel E. It is difficult to know if the FGF_i effects are due to failure of SLUG expression in the medial junctional zone cells as the data is currently presented.
8. Figure 6A shows that apical area is larger in PK1 KD cells, but this is at odds with the mediolateral junction lengths being unchanged in PK1 KD (Figure 1I-J). Further clarity on where in the tissue the mediolateral (ML) lengths were determined is needed, and ML lengths should be quantified in both the medial and lateral junctional zone in the PK1 KD.
9. Some images are pixelated

Version 1:

Reviewer comments:

Reviewer #1

(Remarks to the Author)

This already-thorough manuscript has been substantially improved by the revision. In particular, the orthogonal knockdown using siRNA has removed the only real issue of concern I had with the original submission. I strongly and wholeheartedly support rapid publication of this excellent paper.

Reviewer #2

(Remarks to the Author)

I am very impressed at the responses from the authors to all three reviewers, and I commend them for clarifying any issues we had with the text or data. The added figures and data go above and beyond and dramatically increase the impact of the paper, the ease of understanding, and the overall flow. The images are gorgeous. I feel that the authors responded to all of my concerns and I do not have additional concerns.

Reviewer #3

(Remarks to the Author)

The authors have done an outstanding job responding to the reviewer comments. This is a beautifully executed study with an important message and I recommend publication without delay.

We thank the reviewers for their positive and constructive reviews of our manuscript. Please find below our point-by-point response to the reviewer's questions.

REVIEWER COMMENTS

Reviewer #1 (Remarks to the Author):

I rather like this interesting new paper that not only characterizes an element of neural morphogenesis that has gone all but ignored but finds some interesting new roles for the PCP protein Pk1.

The paper focuses on “junctional neurulation,” the process that is important for linking primary and secondary neurulation in amniote embryos (and for potential clinical relevance) but has received scant attention. Using transgenic quail, the authors very nicely describe the cell behaviors underlying the process, notably showing that apical constriction, EMT, and ingression in the medial region of the neural epithelium are key players. Pk1 had previously been shown to be important for junctional neurulation, and that is confirmed and extended substantially here. Live imaging reveals that Pk1 is essential for apical constriction in these cells. Surprisingly, however, while these cells are planar polarized and seem to be engaged in rosette-mediated convergent extension, pk1 loss does not affect either (but see minor comment 2, below).

I strongly support publication in Nature Communications, both for the detailed characterization of an important but ignored aspect of biology and for the analysis of Pk1. (Indeed, “PCP-independent” roles for PCP proteins are an emerging point of interest.) There is a huge amount of data here and I think it imperative that their publication not be delayed by reviewers who -no doubt- will provide a laundry list of other experiments that might be done. These might even ‘improve’ the paper, but I think it already represents a substantial advance and calls for some intangible “more” should be refused.

I will however make one major comment that must be addressed in some way, as it relates directly to the veracity of the data.

All other comments are mere suggestions.

Major comment:

In the paper, both the Pk1 miRNA and the Slug MO lack specificity controls to rule out off-target effects. The standard in the field for knockdowns in fish and frogs would be to include controls like rescue experiments or phenocopy with an orthogonal reagent (e.g. the same reagent targeting a distinct sequence, or a distinct reagent like a dominant-negative fragment – of which there are many of Pk1).

I do not know the standards in the avian biology community, and it does seem to me that the fly and worm communities use siRNA quite liberally without such controls. But I feel that either off-target controls must be added to the paper, or it must be very explicitly stated in the results section that off-target effects cannot be ruled out.

Although specificity controls are not routinely included in avian knockdown studies, we agree with the reviewer that it is important to address the possibility of off-target effects associated with PK1 knockdown. To this end, we performed an additional independent control experiment using an orthogonal reagent. Specifically, we electroporated Lifeact-EGFP embryos with a previously published siRNA targeting a distinct sequence of PK1 (Dady et al., 2014) or a scrambled control siRNA.

Live imaging overnight revealed that embryos treated with the PK1 siRNA developed defects comparable to those observed following PK1 knockdown by miRNA, including failure of posterior neuropore closure. These data are presented in Supplementary Fig. 1f–i and described in the revised text below:

“Knocking down PK1 expression in the junctional zone disrupted junctional neurulation and prevented closure of the posterior neuropore (Fig. 1k, l, and Supplementary Fig. 1e, Supplementary Movie 3). We found comparable defects following electroporation of an siRNA targeting PK1 (Supplementary Fig. 1f–i). Together, these results confirm that correct formation of the junctional neural tube relies on PK1.”

New data in Supplementary Figure 1 showing PK1 siRNA has comparable effects to PK1 miRNA:

The SLUG morpholino approach has previously been used by three labs to inhibit EMT in avian embryos (Jhingory et al., 2010; Sanchez-Vasquez et al., 2019; Taneyhill et al., 2007). Consistent with these studies, we show that this morpholino significantly reduces the number of SLUG-positive cells in the junctional zone (Fig. 5a and Supplementary Fig. 5a) and disrupts the acquisition of protrusive cell morphology (Fig. 5b, c). As an independent, orthogonal approach to block SLUG expression, we also inhibited FGF signalling, which is required for SLUG expression and cell ingression at the primitive streak in both mouse and chick embryos (Ciruna & Rossant, 2001; Hardy et al., 2011). Inhibition of FGF signalling phenocopied the effects of SLUG knockdown, including reduced SLUG expression and impaired medial cell ingression (Fig. 5f–i).

In addition, we now include new data showing that expression of a dominant-negative FGFR1 construct (Martinez-Morales et al., 2011) in the junctional zone similarly reduces SLUG expression, inhibits medial cell ingression, and results in junctional neural tube defects (Fig. 5l–p and supplementary Fig. 5b). The convergence of phenotypes obtained using distinct molecular perturbations (SLUG morpholino knockdown, pharmacological inhibition of FGF signalling, and dominant-negative FGFR1 expression) strongly supports the conclusion that the observed effects reflect specific disruption of the SLUG-dependent ingression program rather than off-target artefacts.

New data for SLUG MO and dnFGFR1 in Figure 5:

Figure 5

Minor comments:

1. The authors might be more careful/explicit about what they mean by “PCP-independent.” PCP, after all, describes both the signaling pathway and the cellular property. By showing polarization of Vangl remains intact after Pk1 KD, it seems they are right with regards to cellular property. But because they did not disrupt any other PCP protein, Pk1 yet may require the entire signaling pathway to effect ingression.

We agree that it is important to be mindful that PCP refers to both the cellular property and the signalling pathway. We tried immunostaining for other core PCP core proteins including Dishevelled 3 (abclonal, A3842; Bio-Strategy, SC-271295; Bio-Strategy, SC-53819; New England Biolabs, 3218T) and Celsr1 (Sapphire

Bioscience, A96931-50), but unfortunately none of the antibodies we tested work in the quail system. However, our conclusion that planar cell polarity is preserved following PK1 knockdown is supported by quantitative analyses at multiple levels of the PCP hierarchy. Specifically, we quantified the planar-polarized localization of PK1's core PCP binding partner VANGL2 (Fig. 2f), as well as the downstream PCP-associated effector ROCK1 and F-actin itself (Fig. 2g, h). Together, these measurements span the pathway from polarity cue to cytoskeletal remodelling. We have now made this point more explicit in the text as follows:

“However, unexpectedly, PK1 knockdown did not alter the planar polarity of its binding partner VANGL2, or the downstream effector ROCK1 or F-actin itself (Fig. 2f-h).”

We also provide new data showing that PK1 knockdown in the junctional zone does not affect the number or orientation of supracellular actin cables, which are regulated by PCP-dependent signalling in the avian neural tube (Fig. 2i- k) (Nishimura et al., 2012).

New data in Figure 2 showing preservation of supracellular actin cables following PK1 knockdown:

Taken together, these data support the conclusion that both planar cell polarization and PCP signalling outputs are largely preserved following PK1 knockdown. However, we cannot exclude that some aspects of PCP signalling may be altered so we have revised the text to refer more specifically to the cellular property of planar polarization, rather than PCP signalling in a broader sense, where appropriate.

2. Is there really convergent extension in junctional neurulation? Rosettes are shown in Supp Fig. 2D but are not quantified for controls or KDs. The simpler T1 transitions are also not quantified. This oversight is important because the difference in AP/ and ML junction dynamics in Supp. Fig. 2E, F seem barely significant. Is this tissue really doing convergent extension?

A better way to ask the question is to quantify the number of effective event for rosette formation and separately for rosette resolution. As well as the same for T1s. Then ask, does Pk1 change these?

I think these things can be quite easily extracted from the data already in hand and could be important. Such data would help us ask 1) if C?E happens here, 2) if Pk1 controls it and 3) and maybe PCP does control it but without Pk1 (i.e. via Pk2).

[Redacted]

We have examined rosette formation in the junctional zone but concluded that it is not a reliable metric for convergent extension in this system, as rosettes can arise through distinct cellular behaviours, including mediolateral junction shrinkage as well as apical constriction followed by cell ingression (see examples below):

Reviewer Figure 1: Rosettes in the junctional zone are formed both by mediolateral junction contraction and cell ingression.

We have therefore removed the rosette data from Supplementary Figure 2, as it is not quantified and distracts from the central conclusion of Figure 2: that PK1 knockdown does not affect cell convergence towards the midline or the planar polarization of cells within the junctional zone.

[Redacted]

3. In Fig. 1I, it is not clear to me that these images are sufficient to back the claim that there is more Pk1 in these cells. I do not feel that normalizing to actin is sufficient to rule out issues like cell density here. This is important, too, because this assay is the only control to show the Pk1 knockdown is effective. Ideally, I'd like to see a Pk1 western blot to show efficacy of the KD.

We apologise that we did not clearly describe our approach for the PK1 quantitation in the main text. Embryos were immunolabelled for PK1 and N-cadherin and stained with Phalloidin and DAPI to provide morphological landmarks. We used the N-cadherin signal to computationally segment each cell and then the PK1 intensity was measured per cell. The PK1 intensity in the junctional zone was then normalised to the intensity in the primary neural tube for each embryo, to avoid variation between individual embryos. The individual and mean data shown in the quantitation in Fig1i show that although there is a broad range of intensities, there is significantly more PK1 in the junctional zone than the primary neural tube at the 5 somite stage and the 9 somite stage.

We have stated our approach more explicitly in the text as follows:

“To examine PK1 expression, we performed immunofluorescent staining. Cells were computationally segmented using an N-cadherin signal and the PK1 intensity was measured per cell. Our analyses showed expression of PK1 was significantly enriched in the junctional zone compared to the primary neural tube, from the 5ss to the 9-10ss (Fig. 1i, j).”

We have also provided additional images showing PK1 alone in black and white, PK1 with DAPI, PK1 with Phalloidin (Fig. 1i) and PK1 with N-cad (Fig. 1j).

New data in Figure 1 showing PK1 expression in the junctional zone:

While we agree a PK1 western blot would be an excellent way to confirm expression levels, unfortunately this approach is not feasible due to the extremely small amount of material in the junctional zone of each embryo. However, we are confident the PK1 knockdown is effective as we see 1) reduced immunostaining for PK1 (Supplementary Fig. 1d), 2) the expected defects using both PK1 miRNA and PK1 siRNA (see new data Supplementary Fig. 1f – i), and 3) these results are in accordance with previous work knocking down PK1 in the avian neural tube (Dady et al., 2014).

4. How is it possible the distributions in fig. 2C are so hugely significant, but the aggregate difference in fig. 2dis barely significant? I am confused.

2c is the pooled result of all cells from 3 embryos (n = 806 cells for control and 796 cells for PK1 KD) whereas d shows all individual cells and the mean for each embryo. In d the statistical analysis compares the means of the embryos.

5. The literature in this field is vast and complex. I hope the reviewers will consider the following suggestions concerning their treatment of it.

5a. Citation 17 (Blankenship) is inappropriate. That paper says nothing of PCP proteins and indeed deals with *Drosophila* CE, which is independent of PCP proteins.

This citation has been removed.

5b. Citation 19 is fine, but this connection between PCP, CE, and neural tube closure was shown first by Wallingford and Harland ten years prior (Development 2002).

This citation has been added.

5c. It seems a shame the Discussion does not consider the several papers from the Sokol Lab on PCP in apical constriction during neurulation (e.g. papers by Ossipova in 2014 and 2015; there are others).

5d. Likewise, the curious results here are lent credence by recent work from the Devenport lab on PCP-independent roles for Vangl2 (papers last year from Paramore in Dev Cell and Development).

5e. It is relevant to this paper's findings that while Pk1 was linked to C/E in frogs, fish, and ascidians, the phenotype of the Pk1 mouse mutant was a defect not in PCP but in apical/basal polarity (Tao et al., 2009).

We thank the reviewer for reminding us of these papers which are highly relevant to our manuscript. We have now included this work in our revised Discussion as follows:

“ PK1 is broadly expressed throughout vertebrate embryogenesis and studies in mouse, Xenopus, zebrafish and avian embryos have demonstrated its requirement in neurulation and the development of cardiac, respiratory, renal, skeletal, dental, auditory, and ocular structures (Radaszkiewicz et al., 2024). In frogs, fish, and ascidians, PK1 has been closely linked to convergent extension movements, reinforcing its classification as a core component of the PCP pathway. Accordingly, disruptions in these diverse developmental processes have typically been attributed to defects in planar cell polarity. However, analysis of the Pk1 mouse mutant revealed a distinct phenotype characterized by defects in apicobasal polarity rather than in PCP, indicating that PK1 can function outside of canonical planar polarity (Tao et al., 2009). Consistent with this, during Xenopus neurulation, PCP proteins including Prickle and Vangl are required for apical actomyosin accumulation and neural plate bending, implicating PCP components directly in the regulation of apical constriction rather than exclusively in mediolateral cell intercalation (Ossipova et al., 2015; Ossipova et al., 2014). More recent work has provided compelling evidence for PCP-independent functions of Vangl2 in epithelial organization and tissue morphogenesis (Paramore, Goodwin, et al., 2024; Paramore, Trenado-Yuste, et al., 2024). Together with growing evidence of PCP protein functions independent of planar polarity (Huang & Winklbauer, 2022; Ranie & White, 2025; Yoon et al., 2023), these findings support an emerging view that PK1 promotes apical constriction as a key morphogenetic process. Our data extend this paradigm by demonstrating that PK1 functions independently of planar cell polarization in the junctional zone, where it drives apical constriction and medial cell ingression, suggesting that PK1 may regulate actomyosin-dependent epithelial remodeling in multiple developmental contexts beyond junctional neurulation.”

5f. Ref 6 seems to be missing the journal name and page numbers, etc.

This reference has been corrected.

Reviewer #2 (Remarks to the Author):

Summary: The manuscript, “Quantitative live imaging reveals PRICKLE1 controls junctional neural tube morphogenesis independent of Planar Cell Polarity,” focuses on understanding the mechanisms driving the closure of the neural tube between the primary and secondary neurulation regions in the spinal cord of transgenic quail embryos. They identify that the PCP protein, Prickle1 (PK1), drives apical constriction via actomyosin accumulation and that it is necessary for junctional neural tube closure at the border where mediolateral convergence in primary neurulation and cell ingression during secondary neurulation occur. Overall, this is a very interesting paper, and the text conclusions are well supported by the data. My comments are below.

Comments:

1. In the introduction do the authors mean amniotes when they refer to higher vertebrates? If so, it might make it clearer to state that because fish and amphibian neurulation processes may differ from those described here.

We appreciate that our wording may have caused confusion and have changed it as follows to be more accurate:

“Neurulation is a critical early embryonic process in vertebrates which forms the neural tube, the precursor to the brain and spinal cord (Greene & Copp, 2014). Failures in neurulation cause severe congenital malformations called neural tube defects (NTDs) which are amongst the most common human birth defects. In amniotes, the neural tube forms from head-to-tail, through two fundamentally different morphogenetic processes (Greene & Copp, 2014).”

2. Can the authors provide a black and white image of PK1 alone in addition to the overlay with DAPI in Figure 1? Adding a PK1 with Phalloidin overlay would be helpful here too.

We have now provided a black and white image of PK1 immunostaining alone and images of PK1 with DAPI and PK1 with Phalloidin in Figure 1i. We have also added images of PK1 and N-cadherin immunostaining in the primary neural tube and junctional zone in Fig1j.

New data in Figure 1 showing PK1 expression in the junctional zone:

3. I can see that the knockdown cells still move significantly less, but the cell convergence tracks in the control embryo in figure 2B are very different (less convergence) than the tracks in figure 1E, which seem very directed to the midline. Are these embryos different stages or is there an explanation for this difference?

The reviewer is correct that these embryos are different stages. The embryo shown in Fig 1e was tracked from the 5 somite stage as this is the beginning of junctional neurulation. The embryo in Figure 2b was tracked from the 7 somite stage to ensure there was sufficient time for PK1 expression to be disrupted following electroporation of the PK1 miRNA (or scrambled control) construct.

4. The significance indicators (asterisks) are missing from the graph in figure 2C that shows displacement.

We thank the reviewer for drawing our attention to this omission which has now been corrected.

5. I like that in figure 4, the rationale and experimental design is great and supports the foundation that

these cells are ingressing using traditional EMT mechanisms. Although this may be out of scope for this paper, it would be interesting to see how the localization of cell adhesion molecules changes in these cells as the adherens junctions are remodeled during PK1-mediated ingression. At these early stages E-cadherin is expressed in the neural ectoderm while N-cadherin is expressed in the underlying mesenchymal cells (Dady et al., 2012 and Rogers et al., 2018) and slug (in theory) may be linked to driving these changes to promote EMT. I would like to see cadherin localization changes in the PK1 knockdowns and the Slug knockdowns (figure 5 and figure 7). Also- are the adherens junctions intact in these embryos where the cells do not ingress (maintenance of cell-cell adhesion and lack of constriction)? Is the basement membrane still remodeled/removed in the midline although the cells cannot navigate in?

We thank the reviewer for highlighting the potential role of cadherin remodeling during PK1-mediated ingression. While cadherin switching is often associated with EMT in many systems, its role during junctional neurulation appears to differ from that described for the primary neural tube.

Although early neuroectodermal cells of the primary neural tube express E-cadherin, their subsequent switch to N-cadherin is thought to reflect neural commitment rather than EMT (Dady et al., 2012). In addition, cells in more caudal regions express N-cadherin prior to formation of the secondary neural tube. Consistent with this, we now provide new data showing that, at the onset of junctional neurulation, cells in the junctional zone already resemble the secondary neural tube, expressing N-cadherin and the neural marker SOX2 (Fig. 1c), but not E-cadherin (Supplementary Fig. 4a).

New data in Supplementary Fig 4 showing expression of E-cadherin and N-cadherin in the junctional zone:

We have updated the text as follows:

“Cells in the primitive streak undergo an epithelial-to-mesenchymal transition (EMT) during which they constrict their apical surfaces, ingress into the deeper tissue layer and switch from expression of the epithelial cadherin E-cadherin to the mesenchymal cadherin, N-cadherin (Francou et al., 2023; Nakaya et al., 2008). Therefore, we hypothesized that a similar mechanism may drive the ingression of cells in the medial region of the junctional zone.

The SNAIL gene family member, SLUG, is essential for EMT and cell ingression at the primitive streak (Nieto et al., 1994). Immunofluorescent staining revealed that cells in the junctional zone already express N-cadherin and not E-cadherin (Supplementary Fig. 4a), but many cells localized to the medial region were SLUG⁺ (Fig. 4a).”

In response to the reviewer’s request, we performed immunostaining for both E-cadherin and N-cadherin following PK1 knockdown. We did not observe any changes in either the expression or localization of these cadherins. All cells within the junctional zone expressed N-cadherin and not E-cadherin, regardless of whether they underwent ingression.

Reviewer Figure 2: PK1 knockdown does not alter the expression of E-cadherin or N-cadherin in the junctional zone.

As these findings were negative, they were not included in the current manuscript; however, they can be added if the reviewer considers it to be necessary.

We did not perform cadherin staining in SLUG knockdown embryos, as our data indicate that cadherin expression is uniform across the junctional zone and not correlated with SLUG expression or ingression status (Figure 1j):

Reviewer Figure 3: E-cadherin is not expressed in SLUG-positive cells in the junctional zone.

Moreover, SLUG is classically associated with transcriptional regulation of cadherins in contexts where cadherin switching occurs; however, because junctional zone cells already express N-cadherin prior to ingress, we reasoned that SLUG is unlikely to regulate EMT in this system through modulation of cadherin identity. Instead, our data support a model in which PK1- and SLUG-dependent ingress occurs without classical cadherin switching or gross disruption of adherens junctions. These observations suggest that the failure of cells to ingress in PK1-deficient embryos is not due to inappropriate cadherin expression.

We do have evidence that basement membrane/ECM organization is altered at the junctional neural tube when PK1 function is disrupted. In particular, Fibronectin becomes disorganized in PK1-deficient embryos, consistent with previous observations (Dady et al., 2014). However, this disruption occurs at stages later than those used for our live-imaging analyses and therefore does not directly coincide with the initial failure of cell ingress described in this study.

Reviewer Figure 4: PK1 knockdown disrupts Fibronectin organisation in the junctional neural tube after the 20ss.

This finding is also consistent with the known role of Prickle proteins in regulating extracellular matrix deposition and organization in other systems, including zebrafish, mouse and *Xenopus* (Dohn et al., 2013; Goto et al., 2005; Tao et al., 2009). While these data suggest that PK1 may contribute to later basement membrane remodelling, a detailed analysis of the basement membrane and its relationship to failed cell ingressión would require additional experiments beyond the scope of the current manuscript.

Altogether, given the absence of cadherin switching, we propose that this ingressión process is more accurately described as a partial or hybrid EMT rather than a “traditional” EMT, and we have revised the text accordingly:

“Together, these findings demonstrate that cells in the junctional zone of the neural plate undergo at least a partial EMT once they reach the medial region to ingress towards the ventral side.”

6. In figure 4G, the authors use fibronectin as a marker of the basement membrane, but that is not a usual marker of the basement membrane itself like laminin for example. They may want to alter the text here as fibronectin can be organized by the existing basement membrane but this would be an indirect readout.

We agree with the reviewer and have provided additional data showing that Laminin localisation is very similar to Fibronectin in the primary neural tube and junctional zone (Supplementary Fig. 4e). Our results agree with previous work showing a Laminin-free zone located caudo-medially to Hensen’s node in the 6 somite stage chick embryo (Shimokita & Takahashi, 2011). Importantly, both Laminin and Fibronectin are absent from the medial region of the junctional zone where cells are ingressing.

New data in Supplementary Figure 4 showing Laminin is absent from the medial region of the junctional zone:

We have updated the text as follows:

“Another crucial early step in EMT in the primitive streak is degradation of the basement membrane (Nakaya et al., 2008). During primary neurulation, the ventral side of the neural plate is covered by a layer of the extracellular matrix proteins Fibronectin (FN) and Laminin (Fig. 4f and Supplementary Fig. 4e, top). This layer remains intact in the lateral regions of the junctional zone but is absent in the medial region, consistent with EMT-driven medial cell ingressión (Fig. 4f and Supplementary Fig. 4e, bottom).”

7. In my opinion, the FGF blocking experiments don’t really add much to the story and in fact- they may complicate it a bit. The Stern, Harland, and Bronner labs (among others) have shown that FGF signaling is required for neural plate and neural plate border induction and it is not clear that in the embryos treated

with the FGF inhibitor (figure 5 D-H) that these tissues are specified appropriately. I would either remove these experiments, or I would show that SOX2 and neural plate border (PAX7) expression is normal in the embryos treated with the FGF inhibitors.

We appreciate the reviewer's insightful comment regarding the role of FGF signalling in neural plate and neural plate border induction. However, we wish to clarify that in our experiments, embryos were treated with the FGF inhibitor only from the 1-somite stage, which is after neural plate and neural plate border induction has already occurred. In avian embryos, neural plate and border induction occurs during gastrulation and is largely established by HH3–HH4, prior to somite formation (Roellig et al., 2017; Stern, 2005). Therefore, treatment at the 1-somite stage should not disrupt the initial specification of these tissues, but instead primarily affects later processes such as cell ingression and morphogenetic movements, which are the focus of our study.

We feel that the FGF inhibition experiments are important to demonstrate that the medial cells in the junctional zone are undergoing a partial EMT, and that this is required for junctional neural tube formation. We have now strengthened our findings with additional data showing that inhibition of FGF signalling (either with an inhibitor or a dominant-negative FGFR1 construct) reduces SLUG positive cells in the junctional zone, and blocking FGF specifically disrupts junctional neurulation (Fig. 5k, p and Supplementary Fig 5c).

The text has been modified as follows:

“To confirm that blocking FGF signalling reduced SLUG expression, we fixed embryos at the 5ss when junctional neurulation initiates and stained for SLUG. In the control embryos, 74.70 ± 3.09 % (mean \pm sem) of cells in the medial junctional zone were SLUG⁺, whereas this reduced to 51.51 ± 4.93 % in the FGF-inhibited embryos (Fig. 5f, g). Consistent with the effects of reducing SLUG expression, in the FGF-inhibited embryos the H2B-EGFP labelled cells remained significantly closer to the dorsal surface of the junctional zone than in the DMSO control embryos and no cells penetrated to the ventral third of the tissue (Fig. 5h, i). Furthermore, by the 9-10ss the posterior neuropore had typically closed in control embryos with an average width of 45.40 ± 17.27 μ m. By contrast, the posterior neuropore remained open in FGF-inhibited embryos (average width of 183.50 ± 8.91 μ m), resulting in a junctional neural tube defect (Fig. 5j, k). To exclude potential confounding effects of FGF inhibition in surrounding tissues, we electroporated plasmids encoding a dominant-negative FGFR1 (Martinez-Morales et al., 2011) into the neural tube of 1ss wildtype quail embryos. This more targeted inhibition also reduced the number of SLUG⁺ cells in the junctional zone, inhibited the ingression of medial cells and resulted in junctional neural tube defects characterised by an open posterior neuropore (Fig. 5l – p and Supplementary Fig. 5b). These findings confirm that FGF-dependent SLUG expression is required in the junctional zone to drive ingression of the medial cells and enable posterior neuropore closure. Blocking FGF signalling specifically impacted junctional neurulation and did not disrupt the neural tube before the 6ss (Fig. 5k, p) or when the inhibitor was added after PNP closure (Supplementary Fig. 5c). Furthermore, FGF inhibition did not block the assembly of actin cables associated with convergence in the lateral regions of the junctional zone (Supplementary Fig. 5d), suggesting the defect primarily results from the prevention of medial cell ingression in the junctional zone.”

New data in Figure 5 for treatment with SLUG MO and dnFGFR1:

Figure 5

8. In figure 5l the embryo appears torn straight through in the midline rather than a basic failure of neural tube closure that we see in the other figures. It may just be the image provided but this phenotype does not appear to phenocopy the image in figure 1j. Can the authors comment on this or provide a more representative embryo?

The image originally provided was a dorsal view of the treated embryo. We agree that this was not the optimal representation of the junctional neural tube defect induced by azidoblebbistatin, as the defect was partially obscured by the overlying unclosed ectoderm, giving the embryo a torn appearance. We have now

replaced this (and the image of the control embryo) with an X–Y slice of the nuclear mCherry channel taken below the ectoderm, allowing the neural tube structure to be visualized more clearly.

Accordingly, we have now presented images of the embryos treated with the FGF inhibitor and the dominant-negative FGFR1 construct in the same manner to facilitate comparison.

New images of embryos in Figure 5x:

9. I don't think that fibronectin and N-cadherin need to be capitalized in the text, but rather the authors can abbreviate them in the text (FN and N-cad).

We thank the reviewer for this suggestion and have revised the text accordingly.

10. What do the neural ectodermal cells that ingress become? I know figure 1D shows that some of them reincorporate into the neural tube- but how? Do others end up in the mesodermally-derived tissues? I would like to see a section of the embryo from 1D to demonstrate this. For those cells that don't reincorporate, do they die after ingress? Apoptosis at junction points is necessary for neural tube closure (Roellig et al., 2022) so it would be interesting to look at activated caspase or TUNEL in these cells in the wild type and PK1 perturbed embryos to see if these cells are undergoing true ingression or apoptotic mediated anoikis.

We have now included additional images in Supplementary Fig. 1a showing that while most cells labelled in the junctional zone give rise to neuroepithelial cells in the junctional neural tube and secondary neural tube, they also contribute a small number of cells to the caudal somites and remaining labelled cells in the tailbud. The exact mechanisms of how these cells reincorporate into the neural tube later in development is beyond the scope of the current study.

New data in Supplemental Figure 1 showing fate of cells electroporated in the junctional zone:

The text has been adjusted as follows:

“The cells of the junctional zone are SOX2⁺ and give rise to both the junctional neural tube which forms between somites 18-27, many precursors for the secondary neural tube initiating from somite 28, a few cells in the caudal somites and cells in the tailbud (Fig. 1c, d and Supplementary Fig. 1a).”

Apoptosis is indeed known to occur at the dorsolateral hinge points of the primary neural tube, where it facilitates neural tube bending (Roellig et al., 2022). Consistent with this, we detect robust activated Caspase-3 labelling in the primary neural tube. However, the junctional zone does not form a medial hinge point, and we observe very few Caspase-3–positive cells in this region. Furthermore, unlike the localization of SLUG, pMLC or PK1, the few caspase-positive cells we see in the junctional zone are not enriched in the medial region.

Together, these results indicate that apoptosis is unlikely to play a major role in medial cell clearance during junctional neurulation.

Reviewer Figure 5: Few cells are positive for activated Caspase-3 in the junctional zone.

We are confident that medial cells undergo true ingression rather than apoptotic removal, as supported by multiple independent lines of evidence. Live-cell tracking demonstrates that, following image registration, medial cells actively move ventrally relative to a stationary dorsal tissue surface (Fig. 3b and Supplementary Fig. 3b,c). In dorsal views (Fig. 3f and Supplementary Movie 5), these cells undergo apical constriction and subsequently disappear from the superficial epithelial layer, consistent with active ingression rather than extrusion of dying cells.

Furthermore, medial cell ingression is accompanied by classical features of EMT, including SLUG expression, morphological changes, and loss of basement membrane (Figs. 4 and 5). Importantly, inhibition of SLUG expression blocks medial cell ingression and results in junctional neural tube defects, supporting a requirement for EMT-associated mechanisms in this process.

We have now performed targeted inhibition of actomyosin contractility using photoactivation of azidoblebbistatin specifically in the medial region. We show this manipulation selectively inhibits medial cell ingression, which has the secondary effect of reducing lateral cell convergence (Fig. 5q – v). Our findings suggest that the main defect of PK1 knockdown is the impairment of medial cell ingression and the continued presence of the medial cells reduces the space available for lateral cells to converge to the midline. These findings are inconsistent with PK1 KD causing apoptotic-mediated anoikis, where dying cells would normally be extruded and removed from the tissue.

New data in Figure 5 showing that photoactivation of azidoblebbistatin in the medial region inhibits lateral cell convergence:

11. If there is space it would be nice to move Supp. Figure 3C to the main body of the paper. The morphology of the slug positive neuroepithelial lateral cell vs. the mesenchymal ingressing medial cell are very cool and demonstrate the differences in these regions really well.

We thank the reviewer for this suggestion and have moved the supplemental figure to the main Figure 4e.

Reviewer #3 (Remarks to the Author):

In this paper, Wang et al investigate the mechanisms underlying neural tube closure in an understudied region of the neural tube – the junctional zone – where primary neurulation and secondary neurulation meet. Using live and quantitative imaging of transgenic quail embryos they show that the junctional neural tube forms through mediolateral convergence and EMT-driven ingression, dependent on the PCP component PRICKLE1. They further present data suggesting that PK1 acts in a PCP-independent manner at the apical cortex of cells, where it drives actomyosin accumulation and apical constriction.

Overall, these findings are significant because they identify both a new cellular behavior critical for neural tube closure and the underlying molecular mechanism, in an emerging model system that is relevant for understanding human disease. They also define new and unexpected functions for PK1, a gene that is implicated in neural tube defects in humans. The authors support their conclusions with robustly quantified data generated through advanced live imaging techniques. The data is presented clearly, and most of the conclusions are consistent with their data. Notably, this work is done in a technically state-of-the-art system of transgenic quail embryos, with similarities to human neural tube formation not present in mouse. For these reasons this paper would be of broad interest to many, including cell biologists, developmental biologists focused on morphogenesis, and clinicians interested in structural birth defects.

Some conclusions, however, could be toned down or strengthened by additional evidence. A main conclusion of the paper is that PK1 acts independently of PCP in junctional neurulation, but the evidence is

rather weak and would require additional controls and experimental data. Toning down this conclusion would also alleviate this concern, as I feel it is a minor point and wouldn't take away from the overall novelty and advance of the paper.

Major comments:

1. One point that needs further investigation regards the uncoupling of PK1 activity from classical PCP axis establishment. The authors conclude that PK1 functions independently of planar polarity cues by assessing VANGL2 localization as well as the length of mediolateral junctions. However without knowing how loss of Vangl2 affects neurulation and junction length, this data is hard to interpret. Another possibility is that PK1 is required downstream of Vangl2. Does VANGL2 knockdown result in a difference in length of mediolaterally oriented junctions? This should be added to justify the use of mediolateral junction length as a metric for planar polarity function in the junctional zone. Is PK1 localization also VANGL2 independent?

Our conclusion that planar cell polarization is preserved following PK1 knockdown is supported by quantitative analyses at multiple levels of the PCP hierarchy. Specifically, we quantified the planar-polarized localization of PK1's core PCP binding partner VANGL2 (Fig. 2f), as well as the downstream PCP-associated effector ROCK1 and F-actin itself (Fig. 2g, h). Together, these measurements span the pathway from polarity cue to cytoskeletal remodelling. We have now made this point more explicit in the text as follows:

“However, unexpectedly, PK1 knockdown did not alter the planar polarity of its binding partner VANGL2, or the downstream effector ROCK1 or F-actin itself (Fig. 2f-h).”

Mediolateral junction length has previously been used to measure planar polarity in *Xenopus* (Christodoulou & Skourides, 2022; Shindo & Wallingford, 2014). However, we have now reanalysed our imaging data to quantify the number and orientation of supracellular actin cables, which are known to depend on PCP signalling in the avian neural tube (Nishimura et al., 2012). Our new data show that PK1 knockdown does not affect either the abundance or orientation of these supracellular actin cables (Fig. 2i–k), as now described in the text:

“Furthermore, PK1 knockdown in the junctional zone had no effect on the number or orientation of supracellular actin cables, which are regulated by PCP-dependent signalling in the avian neural tube (Fig. 2i- k)(Nishimura et al., 2012).”

New data in Figure 2 showing preservation of supracellular actin cables following PK1 knockdown:

Taken together, these complementary analyses indicate that establishment of the classical PCP axis and its downstream cytoskeletal outputs are largely preserved following PK1 knockdown. To avoid ambiguity, we have revised the manuscript to refer more precisely to the cellular property of planar polarization, rather than PCP signalling in a broader sense, where appropriate.

2. Can the authors elaborate about the relationship between EMT/ingression and mediolateral convergence? Does EMT-driven ingression of medial neuroepithelial cells contribute to the mediolateral convergence of lateral cells? Both processes are affected by PK1 KD, but are these two separate behaviors driven by two PK1-dependent processes or is convergence a passive consequence of ingression? Can the SLUG KD data be used to explore how inhibiting EMT affects mediolateral convergence? Alternatively, could the authors use azidoblebbistatin to inhibit myosin selectively in the medial zone and determine the impact on lateral cell movements?

We thank the reviewer for this question which led to data that have strengthened our results and clarified the conclusions. We repeated the azidoblebbistatin experiments and used live cell tracking to examine the impact on lateral convergence. As expected, photoactivation of azidoblebbistatin in the medial region inhibited the ingression of medial cells (Fig. 5r, s). Interestingly, this also resulted in a reduction of lateral cell convergence (Fig. 5t – v). These results suggest that the primary defect arising from PK1 disruption is the inhibition of medial cell ingression. Impaired lateral cell movement is likely a secondary effect of failed medial cell clearance, which limits the space available for lateral cells to converge to the midline.

Our new data is described as follows:

“Prior to ingression, the medial cells undergo apical constriction (Fig. 3f, g), a process that depends on actomyosin contraction (Martin & Goldstein, 2014). Therefore, to selectively target cell ingression in the medial region of the junctional zone, we used a photo-activatable myosin II inhibitor, azidoblebbistatin (Kepiro et al., 2012; Zenker et al., 2018). First, we electroporated 1ss Lifeact-EGFP quail embryos with a plasmid encoding H2B-RFP into the junctional zone to facilitate cell tracking. Next, we applied azidoblebbistatin at the 5ss and specifically activated it in the medial region using a 405 nm laser (Fig. 5q). Live cell tracking revealed a significant decrease in the ventral displacement of medial cells following photoactivation (from $5.16 \pm 0.47 \mu\text{m}$ to $0.58 \pm 0.11 \mu\text{m}$), confirming that blocking actomyosin contraction inhibited medial cell ingression (Fig. 5r, s). Notably, although myosin II inhibition was restricted to the medial region, lateral cells also exhibited a significant reduction in convergence toward the midline (Fig. 5t–v, from $25.71 \pm 1.59 \mu\text{m}$ to $11.39 \pm 1.82 \mu\text{m}$). These findings suggest that the ingression of medial cells is required to create space that allows subsequent convergence of lateral cells into the midline. To examine the effect at the tissue scale, we treated TgT2(CAG:NLS-mCherry-IRES-GFP-CAAX) embryos at the 5ss with azidoblebbistatin. Consistent with the EMT inhibition, blocking actomyosin contraction in only the medial cells caused focalised junctional NTDs, characterised by a significantly wider posterior neuropore at the ~16ss (Fig. 5w, x). Thus, medial cell ingression is required for the correct formation of the junctional neural tube.”

The discussion has been updated as follows:

“The planar polarization-independent role of PK1 in the junctional zone may explain why PK1 disruptions lead to localized NTDs specifically at the site of neuroepithelial cell ingression, in contrast to mutations in other core PCP proteins such as CELSR1-3 and VANGL1/2 which can impair neural tube closure along the anteroposterior axis (Chen et al., 2018; Juriloff & Harris, 2012; Wang et al., 2019; Wen et al., 2010). We propose that the primary defect following PK1 disruption in the junctional zone is impaired ingression of medial cells. This conclusion is supported by our finding that targeted inhibition of actomyosin contractility confined to the medial region is sufficient to block lateral convergence and induce JNTDs (Fig. 5). Therefore, the defects observed in lateral convergence likely arise as a secondary consequence of the failure to clear medial cells from the junctional zone. Together, our findings not only advance our understanding of the cellular choreography underlying junctional neurulation but also provide mechanistic insight into the etiology of human junctional neural tube defects.”

New data in Figure 5 showing that photoactivation of azidoblebbistatin in the medial region inhibits lateral cell convergence:

3. Data showing PK1 subcellular localization could be improved as it is not clear from the images provided in Figure 7 that it is apical and cortical. The authors should include z-projection images to demonstrate an apical PK1 cap and characterize the subcellular localization of PK1 along the apical-basal axis. Also, what is the evidence that the PK1 antibody used is specific? Additionally, Figure 7B should include a staining for PK1 to demonstrate colocalization to the apical enrichment of F-actin and p-MLC.

We thank the reviewer for highlighting this important point as we did not make our approach clear. The images of PK1 in Figure 7d and f show only the apical surface of the tissue and are not maximum intensity projections through the full cellular depth. Briefly, these images were generated using the Fiji plugin LocalZProjector, to produce a 2D projection encompassing only the apical $\sim 3 \mu\text{m}$ of the tissue, based on the N-cadherin signal (Herbert et al., 2021). Consequently, the PK1 signal shown reflects protein localized specifically at the apical cortex. To further substantiate this apical localization, we have now included additional images of PK1 and N-cadherin immunostaining displaying X-Z views of the 3D image stacks (Supplementary Fig. 7a). In addition, we provide images of a transverse vibratome section through the junctional zone showing PK1 immunostaining and DAPI which corroborate the apical localisation of PK1 observed in the surface projections (Supplementary Fig. 7b).

We have clarified this in the text as follows:

“To characterise PK1 at the apical cortex in the junctional zone, we fixed wildtype quail embryos at the 5ss and immunostained for activated Myosin Light Chain (p-MLC), PK1 and N-cad to label the cell junctions (Fig. 7a, b and Supplementary Fig 7a, b). We used the Fiji plugin LocalZProjector (Herbert et al., 2021) to generate a surface-restricted 2D projection to selectively visualise the apical $\sim 3 \mu\text{m}$ of the tissue, using the N-cad signal to define the apical plane (Galea et al., 2021).”

New data in Supplementary Figure 7 showing apical localization of PK1:

We are confident in the specificity of the PK1 antibody used in this study. The expression pattern observed closely matches the previously reported distribution of *Prickle-1* mRNA (Cooper et al., 2008; Dady et al., 2014). In addition, this antibody has been independently validated and used for immunofluorescence in multiple systems, including avian embryos (Asai et al., 2024; Zhao et al., 2025) and rat (Wang et al., 2025).

We further confirmed that the PK1 antibody does not cross-react with other Prickle isoforms by directly comparing its staining pattern with that obtained using a pan-Prickle antibody (Abcam, ab15577), as shown below.

Reviewer Figure 6: The PK1 antibody does not recognise other isoforms in the quail.

Pan-Prickle + DAPI Pan-Prickle Prickle1 Ab + DAPI Prickle1 Ab + DAPI

We agree that it would be ideal to show colocalization of PK1, pMLC and actin in Figure 7b, however the pMLC and PK1 antibodies are both from the same host species, precluding double-labelling.

4. Does loss of PK1 affect cell fate in the medial junctional zone? Quantification of the number of SLUG+ cells in the medial junctional zone of PK1 KD embryos is needed.

We now show that knocking down PK1 does not change the number of SLUG positive cells in the junctional zone (Supplementary Fig. 6b). The text has been adjusted as follows:

“We fixed and immunostained the embryos for SLUG at 5ss and found that knocking down PK1 did not change SLUG expression levels or the number of SLUG+ cells (Supplementary Fig. 6a - c). This confirms that PK1 does not regulate SLUG expression in the junctional zone.”

New data in Supplementary Figure 6 showing PK1 knockdown does not alter SLUG in the junctional zone:

Other comments:

1. Figure 1l: Include PK1 KD embryos to demonstrate that the PK1 antibody is specific

New data in Supplementary Figure 1 showing the PK1 antibody is specific:

We have now included images of PK1 immunostaining of embryos electroporated with the control miRNA and PK1 miRNA in Supplementary Fig 1d.

2. Figure 1l: Include additional panels to characterize the subcellular localization of PK1

We have provided additional images of PK1 alone or with DAPI, Phalloidin or N-cad staining (Fig. 1i and j) and additional images to confirm apical localisation of PK1 (Supplementary Figure 7a, b).

New images of PK1 expression in Figure 1:

New images showing apical localization of PK1 in Supplementary Figure 7:

3. Figure 2F-G: Planar polarization should be quantified as ML junctional enrichment. Quantifying junctional intensity is not sufficient to demonstrate polarization. ML junctional intensity must be quantified relative to AP junctional intensity to demonstrate planar polarization.

Planar polarisation was measured by quantifying mediolateral junctional intensity relative to anteroposterior intensity as described previously (Christodoulou & Skourides, 2022) and in our Methods section. We thank the reviewer for drawing our attention to the axes on the graphs in Figure 2f-h which should have been labelled Normalized mediolateral junction enrichment (instead of intensity). This has now been corrected.

4. Figure 3B: Can the authors include still frames from the H2BiRFP channel, zoomed in, to demonstrate relative positions of nuclei through time as they ingress? The cell tracks shown don't distinguish between the whole tissue moving ventrally versus individual cells internalizing. This is an important point of the paper and should be shown as convincingly as possible.

We agree that it is essential to demonstrate that the tracking reflects true cell ingress rather than passive tissue drift. We now provide additional data showing that, following image registration, the dorsal surface of the tissue remains stationary and does not move ventrally (Supplementary Fig. 3b), while H2B-mRFP670-labelled cells move ventrally over time relative to the tissue surface (Supplementary Fig. 3c). In addition, the dorsal view shown in Fig. 3f and Supplementary Movie 5 demonstrate that cells constrict their apical surface and subsequently disappear from the superficial layer of the tissue, consistent with active ingress.

New Supplemental Figure 3 showing cells move ventrally, relative to the dorsal surface of the tissue:

5. Figure 3C-J: Suggest removing the lined connected the data points. Lines suggest the x-axis is time, and the data would be clearer without the connections.

We appreciate the reviewer's suggestion regarding the line connections in Figures 3c, j. However, we believe that retaining the lines is important for accurately conveying the structure of the data. The x-axis does not represent time; rather, it represents spatial position across the 300 micrometers width, with each data point corresponding to the mean of cells within a 30 micrometers range. The lines are included to help visualize the

continuous nature of the spatial gradient, rather than to imply temporal progression. Removing the lines would risk obscuring the underlying trend in the data.

6. Figure 5A-C: The authors should validate that the SLUG MO cells in the medial junctional zone that remain dorsal have lost their mesenchymal or protrusive morphology.

We now provide additional data demonstrating that SLUG morpholino (MO)-treated cells lose their protrusive morphology and instead retain an epithelial-like profile (Fig. 5b, c). In response to the reviewer's request, we initially performed experiments in which SLUG or control morpholinos were co-electroporated with a SLUG-enhancer reporter construct, allowing specific visualization of cells in which SLUG expression would normally be active. However, for reasons that remain unclear, the presence of either the SLUG or control morpholino interfered with reporter expression driven by the SLUG enhancer.

We therefore adopted an alternative strategy, co-electroporating the morpholinos with a membrane-RFP construct under the control of the ubiquitous CMV promoter. While this approach successfully labelled cells, the ubiquitous expression of membrane-RFP meant that only a subset of labelled cells (~25–45%) displayed a protrusive morphology in control embryos. Importantly, this proportion was significantly reduced following SLUG knockdown, supporting the conclusion that SLUG is required for the acquisition or maintenance of protrusive behaviour during ingression.

New data in Figure 5 showing SLUG MO-treated cells lose their protrusive morphology:

7. Figure 5D-E: Given that the % of SLUG+ cells is still around 50% in Panel D, include a staining for SLUG expression in the FGFi condition in Panel E. It is difficult to know if the FGFi effects are due to failure of SLUG expression in the medial junctional zone cells as the data is currently presented.

We now include images of the immunostaining for SLUG in the junctional zone in DMSO control and FGFi conditions (Fig. 5g) which show FGF inhibition decreases the number of SLUG-positive cells in the medial region.

New images of SLUG staining in FGF inhibitor-treated embryos in Figure 5:

8. Figure 6A shows that apical area is larger in PK1 KD cells, but this is at odds with the mediolateral junction lengths being unchanged in PK1 KD (Figure 1I-J). Further clarity on where in the tissue the mediolateral (ML) lengths were determined is needed, and ML lengths should be quantified in both the medial and lateral junctional zone in the PK1 KD.

We agree with the reviewer that the original presentation of these data was confusing, and we thank them for highlighting this issue. Mediolateral junction length was initially measured as a readout of planar cell polarity in electroporated cells across the entire junctional zone. As a result, only a small proportion of the analysed cells were located near the midline, where apical constriction and ingression occur. In contrast, measurements of apical area were deliberately restricted to medial cells, as this is the only region in which apical constriction is observed. This difference in sampling contributed to the apparent contradiction in the original presentation.

To address this concern and provide a more appropriate assessment of planar polarity, we have replaced the mediolateral junction length analysis with quantification of the number and orientation of supracellular actin cables, a well-established PCP-dependent feature in the avian neural tube (Nishimura et al., 2012). This revised analysis demonstrates that PK1 knockdown does not alter either the abundance or orientation of these supracellular actin cables (Fig. 2i-k), supporting the conclusion that planar cell polarity is largely preserved.

New data in Figure 2 showing preservation of supracellular actin cables with PK1 knockdown:

9. Some images are pixelated

We thank the reviewer for pointing this out. The apparent pixelation likely arose during the upload or conversion process of the original submission. We have provided the highest-resolution versions available to ensure optimal image quality in the revised manuscript.

References

- Albu, M., Affolter, E., Gentile, A., Xu, Y., Kikhi, K., Howard, S., Kuenne, C., Priya, R., Gunawan, F., & Stainier, D. Y. R. (2024). Distinct mechanisms regulate ventricular and atrial chamber wall formation. *Nature communications*, 15(1), 8159. <https://doi.org/10.1038/s41467-024-52340-3>
- Asai, R., Prakash, V. N., Sinha, S., Prakash, M., & Mikawa, T. (2024). Coupling and uncoupling of midline morphogenesis and cell flow in amniote gastrulation. *eLife*, 12. <https://doi.org/10.7554/eLife.89948>
- Chen, Z., Lei, Y., Cao, X., Zheng, Y., Wang, F., Bao, Y., Peng, R., Finnell, R. H., Zhang, T., & Wang, H. (2018). Genetic analysis of Wnt/PCP genes in neural tube defects. *BMC medical genomics*, 11(1), 38. <https://doi.org/10.1186/s12920-018-0355-9> PMID- 29618362
- Christodoulou, N., & Skourides, P. A. (2022). Distinct spatiotemporal contribution of morphogenetic events and mechanical tissue coupling during *Xenopus* neural tube closure. *Development*, 149(13). <https://doi.org/10.1242/dev.200358>
- Ciruna, B., & Rossant, J. (2001). FGF signaling regulates mesoderm cell fate specification and morphogenetic movement at the primitive streak. *Developmental Cell*, 1(1), 37-49. [https://doi.org/10.1016/s1534-5807\(01\)00017-x](https://doi.org/10.1016/s1534-5807(01)00017-x)
- Cooper, O., Sweetman, D., Wagstaff, L., & Munsterberg, A. (2008). Expression of avian prickle genes during early development and organogenesis. *Dev Dyn*, 237(5), 1442-1448. <https://doi.org/10.1002/dvdy.21490>
- Dady, A., Blavet, C., & Duband, J. L. (2012). Timing and kinetics of E- to N-cadherin switch during neurulation in the avian embryo. *Dev Dyn*, 241(8), 1333-1349. <https://doi.org/10.1002/dvdy.23813>
- Dady, A., Havis, E., Escriou, V., Catala, M., & Duband, J. L. (2014). Junctional Neurulation: A Unique Developmental Program Shaping a Discrete Region of the Spinal Cord Highly Susceptible to Neural Tube Defects. *Journal of Neuroscience*, 34(39), 13208-13221. <https://doi.org/10.1523/jneurosci.1850-14.2014> PMID- 25253865
- Dohn, M. R., Mundell, N. A., Sawyer, L. M., Dunlap, J. A., & Jessen, J. R. (2013). Planar cell polarity proteins differentially regulate extracellular matrix organization and assembly during zebrafish gastrulation. *Developmental biology*, 383(1), 39-51. <https://doi.org/10.1016/j.ydbio.2013.08.027>
- Francou, A., Anderson, K. V., & Hadjantonakis, A. K. (2023). A ratchet-like apical constriction drives cell ingression during the mouse gastrulation EMT. *eLife*, 12. <https://doi.org/10.7554/eLife.84019>
- Galea, G. L., Maniou, E., Edwards, T. J., Marshall, A. R., Ampartzidis, I., Greene, N. D. E., & Copp, A. J. (2021). Cell non-autonomy amplifies disruption of neurulation by mosaic *Vangl2* deletion in mice. *Nature communications*, 12(1), 1159. <https://doi.org/10.1038/s41467-021-21372-4>
- Goto, T., Davidson, L., Asashima, M., & Keller, R. (2005). Planar Cell Polarity Genes Regulate Polarized Extracellular Matrix Deposition during Frog Gastrulation. *Current Biology*, 15(8), 787-793. <https://doi.org/10.1016/j.cub.2005.03.040> PMID- 15854914
- Greene, N. D. E., & Copp, A. J. (2014). Neural Tube Defects. *Annual Review of Neuroscience*, 37(1), 221-242. <https://doi.org/10.1146/annurev-neuro-062012-170354>
- Hardy, K. M., Yatskevych, T. A., Konieczka, J. H., Bobbs, A. S., & Antin, P. B. (2011). FGF signalling through RAS/MAPK and PI3K pathways regulates cell movement and gene expression in the chicken primitive streak without affecting E-cadherin expression. *BMC Developmental Biology*, 11(1), 20. <https://doi.org/10.1186/1471-213X-11-20>
- Herbert, S., Valon, L., Mancini, L., Dray, N., Caldarelli, P., Gros, J., Esposito, E., Shorte, S. L., Bally-Cuif, L., Aulner, N., Levayer, R., & Tinevez, J. Y. (2021). LocalZProjector and DeProj: a toolbox for local 2D projection and accurate morphometrics of large 3D microscopy images. *BMC Biol*, 19(1), 136. <https://doi.org/10.1186/s12915-021-01037-w>
- Huang, Y., & Winklbauer, R. (2022). Cell cortex regulation by the planar cell polarity protein Prickle1. *J Cell Biol*, 221(7). <https://doi.org/10.1083/jcb.202008116>
- Jhingory, S., Wu, C. Y., & Taneyhill, L. A. (2010). Novel insight into the function and regulation of alphaN-catenin by Snail2 during chick neural crest cell migration. *Developmental biology*, 344(2), 896-910. <https://doi.org/10.1016/j.ydbio.2010.06.006>
- Juriloff, D. M., & Harris, M. J. (2012). A consideration of the evidence that genetic defects in planar cell polarity contribute to the etiology of human neural tube defects. *Birth Defects Research Part A: Clinical and Molecular Teratology*, 94(10), 824-840. <https://doi.org/10.1002/bdra.23079> PMID- 23024041
- Kepiro, M., Varkuti, B. H., Bodor, A., Hegyi, G., Drahos, L., Kovacs, M., & Malnasi-Csizmadia, A. (2012). Azidoblebbistatin, a photoreactive myosin inhibitor. *Proc Natl Acad Sci U S A*, 109(24), 9402-9407. <https://doi.org/10.1073/pnas.1202786109>
- Martin, A. C., & Goldstein, B. (2014). Apical constriction: themes and variations on a cellular mechanism driving morphogenesis. *Development*, 141(10), 1987-1998. <https://doi.org/10.1242/dev.102228>

- Martinez-Morales, P. L., Diez del Corral, R., Olivera-Martinez, I., Quiroga, A. C., Das, R. M., Barbas, J. A., Storey, K. G., & Morales, A. V. (2011). FGF and retinoic acid activity gradients control the timing of neural crest cell emigration in the trunk. *J Cell Biol*, 194(3), 489-503. <https://doi.org/10.1083/jcb.201011077>
- Nakaya, Y., Sukowati, E. W., Wu, Y., & Sheng, G. (2008). RhoA and microtubule dynamics control cell-basement membrane interaction in EMT during gastrulation. *Nature Cell Biology*, 10(7), 765-775. <https://doi.org/10.1038/ncb1739>
- Nieto, M. A., Sargent, M. G., Wilkinson, D. G., & Cooke, J. (1994). Control of cell behavior during vertebrate development by Slug, a zinc finger gene. *Science*, 264(5160), 835-839. <https://doi.org/10.1126/science.7513443>
- Nishimura, T., Honda, H., & Takeichi, M. (2012). Planar cell polarity links axes of spatial dynamics in neural-tube closure. *Cell*, 149(5), 1084-1097. <https://doi.org/10.1016/j.cell.2012.04.021>
- Ossipova, O., Chuykin, I., Chu, C. W., & Sokol, S. Y. (2015). Vangl2 cooperates with Rab11 and Myosin V to regulate apical constriction during vertebrate gastrulation. *Development*, 142(1), 99-107. <https://doi.org/10.1242/dev.111161>
- Ossipova, O., Kim, K., Lake, B. B., Itoh, K., Ioannou, A., & Sokol, S. Y. (2014). Role of Rab11 in planar cell polarity and apical constriction during vertebrate neural tube closure. *Nature communications*, 5, 3734. <https://doi.org/10.1038/ncomms4734>
- Paramore, S. V., Goodwin, K., Fowler, E. W., Devenport, D., & Nelson, C. M. (2024). Mesenchymal Vangl1 and Vangl2 facilitate airway elongation and widening independently of the planar cell polarity complex. *Development*, 151(16). <https://doi.org/10.1242/dev.202692>
- Paramore, S. V., Trenado-Yuste, C., Sharan, R., Nelson, C. M., & Devenport, D. (2024). Vangl-dependent mesenchymal thinning shapes the distal lung during murine sacculation. *Developmental Cell*, 59(10), 1302-1316 e1305. <https://doi.org/10.1016/j.devcel.2024.03.010>
- Radaskiewicz, K. A., Sulcova, M., Kohoutkova, E., & Harnos, J. (2024). The role of prickle proteins in vertebrate development and pathology. *Mol Cell Biochem*, 479(5), 1199-1221. <https://doi.org/10.1007/s11010-023-04787-z>
- Ranie, S. N., & White, M. D. (2025). Apical constriction in morphogenesis: From actomyosin architecture to regulatory networks. *Current Opinion in Cell Biology*, 95, 102562. <https://doi.org/10.1016/j.ceb.2025.102562>
- Roellig, D., Tan-Cabugao, J., Esaian, S., & Bronner, M. E. (2017). Dynamic transcriptional signature and cell fate analysis reveals plasticity of individual neural plate border cells. *eLife*, 6. <https://doi.org/10.7554/eLife.21620>
- Roellig, D., Theis, S., Proag, A., Allio, G., Bénazéraf, B., Gros, J., & Suzanne, M. (2022). Force-generating apoptotic cells orchestrate avian neural tube bending. *Developmental Cell*, 57(6), 707-718.e706. <https://doi.org/10.1016/j.devcel.2022.02.020>
- Sanchez-Vasquez, E., Bronner, M. E., & Strobl-Mazzulla, P. H. (2019). Epigenetic inactivation of miR-203 as a key step in neural crest epithelial-to-mesenchymal transition. *Development*, 146(7). <https://doi.org/10.1242/dev.171017>
- Shimokita, E., & Takahashi, Y. (2011). Secondary neurulation: Fate-mapping and gene manipulation of the neural tube in tail bud. *Development, growth & differentiation*, 53(3), 401-410. <https://doi.org/10.1111/j.1440-169x.2011.01260.x>
- Shindo, A., & Wallingford, J. B. (2014). PCP and septins compartmentalize cortical actomyosin to direct collective cell movement. *Science*, 343(6171), 649-652. <https://doi.org/10.1126/science.1243126>
- Shook, D. R., Kasprowicz, E. M., Davidson, L. A., & Keller, R. (2018). Large, long range tensile forces drive convergence during *Xenopus* blastopore closure and body axis elongation. *eLife*, 7. <https://doi.org/10.7554/eLife.26944>
- Shook, D. R., Wen, J. W. H., Rolo, A., O'Hanlon, M., Francica, B., Dobbins, D., Skoglund, P., DeSimone, D. W., Winklbauer, R., & Keller, R. E. (2022). Characterization of convergent thickening, a major convergence force producing morphogenic movement in amphibians. *eLife*, 11. <https://doi.org/10.7554/eLife.57642>
- Stern, C. D. (2005). Neural induction: old problem, new findings, yet more questions. *Development*, 132(9), 2007-2021. <https://doi.org/10.1242/dev.01794>
- Taneyhill, L. A., Coles, E. G., & Bronner-Fraser, M. (2007). Snail2 directly represses cadherin6B during epithelial-to-mesenchymal transitions of the neural crest. *Development*, 134(8), 1481-1490. <https://doi.org/10.1242/dev.02834>
- Tao, H., Suzuki, M., Kiyonari, H., Abe, T., Sasaoka, T., & Ueno, N. (2009). Mouse prickle1, the homolog of a PCP gene, is essential for epiblast apical-basal polarity. *Proceedings of the National Academy of Sciences*, 106(34), 14426-14431. <https://doi.org/10.1073/pnas.0901332106>
- Wang, L., Bu, T., Gao, S., Yun, D., Chen, H., Cheng, C. Y., & Sun, F. (2025). PCP protein Prickle 1 regulates Sertoli cell and testis function via cytoskeletal organization through the recruitment of multiple regulatory proteins. *Am J Physiol Cell Physiol*, 328(6), C2032-C2056. <https://doi.org/10.1152/ajpcell.00861.2024>
- Wang, M., Marco, P., Capra, V., & Kibar, Z. (2019). Update on the Role of the Non-Canonical Wnt/Planar Cell Polarity Pathway in Neural Tube Defects. *Cells*, 8(10). <https://doi.org/10.3390/cells8101198>
- Wen, S., Zhu, H., Lu, W., Mitchell, L. E., Shaw, G. M., Lammer, E. J., & Finnell, R. H. (2010). Planar cell polarity pathway genes and risk for spina bifida. *Am J Med Genet A*, 152A(2), 299-304. <https://doi.org/10.1002/ajmg.a.33230>
- Yoon, J., Sun, J., Lee, M., Hwang, Y.-S., & Daar, I. O. (2023). Wnt4 and ephrinB2 instruct apical constriction via Dishevelled and non-canonical signaling. *Nature communications*, 14(1), 337. <https://doi.org/10.1038/s41467-023-35991-6>
- Zenker, J., White, M. D., Gasnier, M., Alvarez, Y. D., Lim, H. Y. G., Bissiere, S., Biro, M., & Plachta, N. (2018). Expanding Actin Rings Zipper the Mouse Embryo for Blastocyst Formation. *Cell*, 173(3), 776-791 e717. <https://doi.org/10.1016/j.cell.2018.02.035>
- Zhao, Z., Asai, R., & Mikawa, T. (2025). Differential sensitivity of midline development to mitosis during and after primitive streak extension. *Dev Dyn*. <https://doi.org/10.1002/dvdy.70045>